# AI-powered simulation-based inference of a genuinely spatial-stochastic gene regulation model of early mouse embryogenesis

**Michael Alexander Ramirez Sierra** [1,2]*, **Thomas R. Sokolowski** [1]

**1** Frankfurt Institute for Advanced Studies (FIAS), Frankfurt am Main, Germany, **2** Faculty of Computer Science and Mathematics, Goethe-Universität Frankfurt am Main, Frankfurt am Main, Germany

* ramirez-sierra@fias.uni-frankfurt.de

## Abstract

Understanding how multicellular organisms reliably orchestrate cell-fate decisions is a central challenge in developmental biology, particularly in early mammalian development, where tissue-level differentiation arises from seemingly cell-autonomous mechanisms. In this study, we present a multi-scale, spatial-stochastic simulation framework for mouse embryogenesis, focusing on inner cell mass (ICM) differentiation into epiblast (EPI) and primitive endoderm (PRE) at the blastocyst stage. Our framework models key regulatory and tissue-scale interactions in a biophysically realistic fashion, capturing the inherent stochasticity of intracellular gene expression and intercellular signaling, while efficiently simulating these processes by advancing event-driven simulation techniques. Leveraging the power of Simulation-Based Inference (SBI) through the AI-driven Sequential Neural Posterior Estimation (SNPE) algorithm, we conduct a large-scale Bayesian inferential analysis to identify parameter sets that faithfully reproduce experimentally observed features of ICM specification. Our results reveal mechanistic insights into how the combined action of autocrine and paracrine FGF4 signaling coordinates stochastic gene expression at the cellular scale to achieve robust and reproducible ICM patterning at the tissue scale. We further demonstrate that the ICM exhibits a specific time window of sensitivity to exogenous FGF4, enabling lineage proportions to be adjusted based on timing and dosage, thereby extending current experimental findings and providing quantitative predictions for both mutant and wild-type ICM systems. Notably, FGF4 signaling not only ensures correct EPI-PRE lineage proportions but also enhances ICM resilience to perturbations, reducing fate-proportioning errors by 10-20% compared to a purely cell-autonomous system. Additionally, we uncover a surprising role for variability in intracellular initial conditions, showing that high gene-expression heterogeneity can improve both the accuracy and precision of cell-fate proportioning, which remains robust when fewer than 25% of the ICM population experiences perturbed initial conditions. Our work offers a comprehensive, spatial-stochastic description of the biochemical processes driving ICM differentiation and identifies the necessary conditions for its robust unfolding. It also provides a framework for future exploration of similar spatial-stochastic systems in developmental biology.

**Data Availability Statement:** All code files are available from a GitHub repository at https://github.com/MARS-FIAS/Article_Modeling_Mouse_ICM.

git. All estimated parameter posterior distributions are available from the same GitHub repository. A partial (light) data bank is available from a Zenodo repository at https://doi.org/10.5281/zenodo.12637055. The complete (heavy) data bank is split into three distinct Zenodo repositories: https://doi.org/10.5281/zenodo.13891602; https://doi.org/10.5281/zenodo.13896947; https://doi.org/10.5281/zenodo.13896989.

**Funding:** This research work was funded by the LOEWE-Schwerpunkt "Center for Multiscale Modelling in Life Sciences" (CMMS), which is sponsored by the Hessian Ministry of Science and Research, Arts and Culture—Hessisches Ministerium für Wissenschaft und Forschung, Kunst und Kultur—(HMWK). The funders had no role in study design, data collection and analysis, decision to publish, or preparation of the manuscript.

**Competing interests:** The authors have declared that no competing interests exist.

## Author summary

Our study presents a spatial-stochastic model for the gene regulatory network (GRN) and the signaling pathway governing cell-fate differentiation during early mouse embryogenesis, specifically at the blastocyst stage. Departing from biophysics-based models of gene regulation, we perform stochastic simulations of the biochemical processes driving early mouse embryogenesis both at the cell and tissue level. Combining these simulations with state-of-the-art AI-aided inference techniques, we successfully parameterize our model, replicating key experimental observations and providing mechanistic insights into the biochemical interactions giving rise to them. Thanks to the stochastic nature of our approach, we quantify the high robustness of ICM specification to various kinds of noise, and provide quantitative predictions for the effects of diverse experimentally testable perturbations. Altogether, we provide a deeper understanding of the intricate mechanisms driving early cell-fate decisions in mouse embryogenesis, highlighting the synergy of local cellular and broader tissue-scale interactions that shape development.

## Introduction

The process of maintaining cellular plasticity while concurrently directing functional differentiation of individual cells is a cornerstone paradigm in early biological development. A fundamental question connected to this is how a complex multicellular organism can robustly emerge from a single cell, despite the inherent stochasticity or *noise* in the biochemical processes driving cell specification. In early mammalian development, the cell signaling and fate specification dynamics during the preimplantation stage of mouse embryogenesis aid as key catalysts for understanding this core paradigm, serving as an intriguing example of a self-organizing system [1–3].

As such, mouse experimental models have become essential for characterizing genome plasticity, cellular potency, and cell diversity [2, 4, 5], especially considering the rapid advancement of regenerative medicine [6, 7]. It is nowadays relevant for improving stem-cell therapy to unravel the cellular differentiation processes through highly-detailed mechanistic representations of intracellular gene regulation and intercellular signaling, extrapolating from studies of the preimplantation mouse embryo for disentangling human diseases [5–7]. For this, it is particularly important to understand the biochemical mechanisms granting high robustness and reproducibility to early mammalian embryo development [1–3, 7]. It is fascinating that, despite key distinctions in gene-regulatory components among mammal species, the spatio-temporal patterning of early embryos unfolds in a seemingly deterministic fashion at tissue level, while its progression is subject to fundamentally stochastic biochemical processes at single-cell level [1]. In particular, highly robust and reproducible lineage proportioning of the vitally significant cellular-fate transition from inner cell mass (ICM) to epiblast (EPI) and primitive endoderm (PRE) populations relies on neither maternal clues nor other positional information determinants [2]. This process thus correctly proceeds regardless of dissimilar experimental or environmental conditions (in-vivo, ex-vivo, in-vitro, and organoid settings), even while being subject to diverse mechanical or geometrical constraints [2, 8–10].

For the early mouse embryo, these orchestrated processes have been investigated through both experimental and theoretical approaches [1, 11–26], and found to rely on dynamic cross-interactions between gene expression, molecular signaling at the tissue level, plus mechanical cues essential for tissue remodeling and proper cell positioning before implantation [2, 3, 5, 8,

27–37]. The preimplantation stage in mouse embryos involves two pivotal events transforming the zygote into a blastocyst composed of three distinct cell types [13, 29, 30, 38, 39]. Initially, cell segregation results in the formation of the trophectoderm (TE), which contributes to the embryonic part of the placenta, and the ICM, an uncommitted bipotent progenitor tissue which is the source of embryonic stem (ES) cells [40]. Subsequently, a second cell-fate decision within the ICM leads to the differentiation of EPI and PRE tissues. The EPI, marked primarily by NANOG expression, is pluripotent and gives rise to all embryo-proper tissues. In contrast, the PRE, marked primarily by GATA6 expression, contributes to the formation of extraembryonic supporting tissues. Notably, EPI cells secrete FGF4, a signaling ligand that, noteworthily, promotes PRE cell specification. The FGF signaling pathway, which relies on FGF receptors and ERK for downstream transmission, therefore plays a crucial role in regulating the balance between EPI and PRE cells. Interestingly, FGF4 acts within an external feedback loop that ultimately inhibits its own production, downregulating it in PRE cells [8, 21, 41]. The mechanisms underlying the precise and timely distribution of FGF4 across the ICM, critical for appropriate cell lineage specification, remain elusive [9, 42].

For successful embryogenesis, the TE, EPI, and PRE lineages in the blastocyst must be segregated in specific proportions and within a narrow developmental time window [21, 30, 43]. This last specification from ICM to EPI and PRE occurs over approximately 48 hours between embryonic day (E) 2.5 and E4.5 (i.e., 2.5–4.5 days post fertilization) [8, 21]. By E3.0, when blastocyst formation begins, ICM cells co-express NANOG and GATA6. In the further course, EPI and PRE cells emerge asynchronously from ICM cells in a spatially stochastic manner, influenced by the short-range FGF4 signal and regulated by auto- and paracrine feedback loops [9, 30, 44]. A mutually exclusive gene expression profile emerges, with EPI cells exhibiting high NANOG and low GATA6 levels, and vice versa in PRE cells.

This process is driven by a gene-regulatory network characterized by self-activation and mutual repression of NANOG and GATA6, following a common motif for generating multistable expression states [45–49], and is refined by the external FGF4 feedback. The spatio-temporal segregation of EPI and PRE lineages is subsequently coordinated through a mechanical cell-sorting process [37]. The final spatially ordered pattern thus is an emergent property of the differentiating tissue, in stark contrast to systems which employ morphogen gradients for high-precision system-wide coordination of developmental trajectories [50–57], although layers of interacting genes downstream of the gradients can acquire self-organizing and scaling capabilities after their position-dependent activation [49, 58–64].

Leveraging high-throughput single-cell RNA sequencing (scRNA-seq) and similar technologies for quantitative immunofluorescence and genetic profiling [9, 65–70], experimental studies have sought to characterize the dynamics and biophysical parameters of ICM specification, focusing on internal and external developmental cues, molecular mechanisms underpinning gene transcription and mRNA translation, and key signaling pathway components [17, 44, 71–73]. Nevertheless, this remains challenging, due to the typically high sparsity and low granularity of scRNA-seq and similar data [74–77], as well as the low mRNA and protein abundance of relevant biochemical species, leading to significant biological noise. Here we categorize biological noise as either intrinsic (related to the discrete and stochastic nature of biochemical reactions and molecular transport within cells) or extrinsic (referring to external fluctuations affecting all cells). This noise strongly impacts the reliability of developmental dynamics, particularly during early development [78–84], and at the same time hampers experimental measurements.

For these reasons, mathematical and computational models have established themselves as potent tools capable of providing mechanistic insights into biochemical noise control strategies. However, existing models of EPI-PRE specification employ primarily deterministic

methods, treating noise as a secondary feature [11, 15, 22, 23, 38]. These models do not fully capture the stochastic nature of transcription, translation, and signaling, among other processes, and therefore may not accurately reflect the impact of biochemical noise on early embryo development.

More recently, deep-learning-based approaches have been successfully used for both pattern recognition and reconstruction of ICM organoid data, employing experimental and synthetic datasets for training [85]. Although these approaches have good predictive power for both cell-fate spatial composition and determination, they do not reveal full mechanistic insights into the gene regulatory interactions that orchestrate ICM patterning.

In this study, we investigate cell specification during blastocyst formation in the early mouse embryo under truly stochastic conditions. In order to elucidate mechanisms that enable robust and replicable cell-type proportioning in the ICM, we have constructed a spatial model that allows for exact simulation of the stochastic dynamics of key biochemical species in the developing ICM. This model incorporates three distinct diffusive-signaling modes between neighboring cells: autocrine, paracrine, and intermembrane ligand-exchange (akin to juxtacrine) signaling via FGF4. As such, our focus is the gene regulatory and cell-cell signaling processes coordinating the differentiation and proportioning of mouse ICM derived lineages (EPI-PRE). We bypass any representation of force interactions between cells (i.e., exclude cell division, proliferation, motility, and sorting), for properly quantifying the synergistic effects of stochastic biochemical processes working at multiple scales without the influence of extrinsic variability.

Departing from traditional event-driven algorithms for simulating the Reaction-Diffusion Master Equation (RDME) in a spatial context, we devised a scheme for simulating the stochastic evolution of the core biochemical species governing early ICM development at the cellular and tissue scales. Our approach leverages the Simulation-Based Inference (SBI) framework, combining our spatial-stochastic simulator with an advanced AI-based inference technique: the Sequential Neural Posterior Estimation (SNPE) algorithm [86, 87]. This allowed us to perform millions of individual stochastic simulations of our spatial system and to use these data to infer parameter sets that align with desired system behavior.

By integrating our biochemically realistic spatial simulations with SBI, we learned parameters that accurately replicate key experimental observations of mouse ICM cell-lineage differentiation, including its temporal dynamics based on reported lifetimes of the involved biochemical species. Successful parametrization of our explicitly stochastic model enabled us to explore the effects of various sources and degrees of system perturbations, providing insights into the potential role of noise in the functional development of mouse blastocyst cell populations, and predicting system properties that so far remained unaddressed.

Our findings predict point and interval estimates of biophysical model parameters, while suggesting that the ICM system, being naturally biased towards a default "raw" state, requires tissue-level coordination for fate differentiation. Specifically, we recapitulate that: (1) early emergence of the EPI lineage is essential for stimulating the PRE fate; (2) the target cell-lineage proportions arise independently of the number of cells in the system; (3) ICM plasticity and its sensitivity to exogenous FGF4 are coordinated in time. In addition, we predict that: (4) intercellular communication enhances the robustness of ICM fate differentiation compared to a purely cell-autonomous process; (5) moderate variability in cellular initial conditions can enhance the accuracy of establishing the proper cell-fate ratio. These recapitulations and predictions provide quantitative bases for experimentally testing the hypotheses and insights contributed by our computational approach.

Our study not only quantitatively explains how the early mouse embryo's resilience to biological noise arises from the FGF4-driven tissue-level coordination mechanism, but also

underscores the existence of critical windows in developmental timing that dictate cell plasticity and responsiveness to neighboring signaling molecules. On the technical and conceptual side, it exemplifies that AI-driven simulation-based inference can be instrumental in uncovering mechanistic details of system-wide coordination of noise control in highly stochastic biophysical systems.

## Results

### Multiscale spatial-stochastic model of mouse blastocyst development: from ICM progeny to EPI and PRE lineages

We constructed a biophysics-based model of mouse embryo blastocyst formation that features a stochastic-mechanistic description of its intracellular gene-regulatory network and its intercellular diffusion-based signaling processes. It focuses on the differentiation of epiblast (EPI) and primitive endoderm (PRE) lineages from the inner cell-mass (ICM) progenitor population. This differentiation is driven by mutual regulatory interactions between the NANOG and GATA6 genes which act as primary markers of the EPI and PRE fates, respectively. It also accounts for their interactions with FGF4 via FGF receptors and the ERK signaling cascade, which implements a sensing mechanism for external FGF4; the FGF4 proteins are secreted by cells that commit to the EPI fate. We describe our model and simulation methods in detail in section "Computational model of mouse blastocyst (ICM cell differentiation)" of the Methods part.

For computational feasibility, our model represents the developing tissue as a static two-dimensional (2D) lattice, mimicking a monolayer or 2D cellular culture [9]. This structural representation preserves the essential characteristic of the mouse blastocyst as a densely packed assembly of pluripotent cells that can spatially communicate among each other. Similar geometric assumptions have been successfully utilized in previous ICM models [15, 22, 23]. Each voxel of the lattice symbolizes an individual embryonic cell, providing an accessible foundation to analyze cell-cell communication modes (for more details, see "Model at tissue scale").

Derived from established Reaction-Diffusion Master Equation (RDME) simulation schemes, our approach computes realistic molecular count time series using accurate lifetimes for essential biochemical species (detailed in "Model at cell scale"). This results in an authentic reproduction of the temporal dynamics of ICM fate specification, as seen in Fig 1. We opted for an event-driven scheme in order to maximize computational efficiency of our simulations, which is a necessary precondition for applying the Simulation-Based Inference (SBI) framework to it, as described next.

### Exploring model parameter space via SBI

In computational modeling of complex biological systems, inferring parameter sets that reproduce experimental observations is a key challenge. This usually requires specific domain knowledge, as the selection of an appropriate parameter search method is chiefly influenced by the particular properties of the problem at hand [88]. One significant hurdle in developmental biology is the lack of a versatile, general-purpose inference method suitable for high-dimensional stochastic models, which typically require large and comprehensive datasets with fine-grained resolution. This expands both in terms of the features measured (e.g., gene expression levels at the single-cell scale rather than bulk measurements) and the temporal resolution (e.g., time-series data capturing dynamics at all relevant scales). However, most experimental studies are unable to simultaneously measure all system variables required for such in-depth mechanistic representation. To bridge this gap, Biologically-Informed Neural Networks (BINNs) and

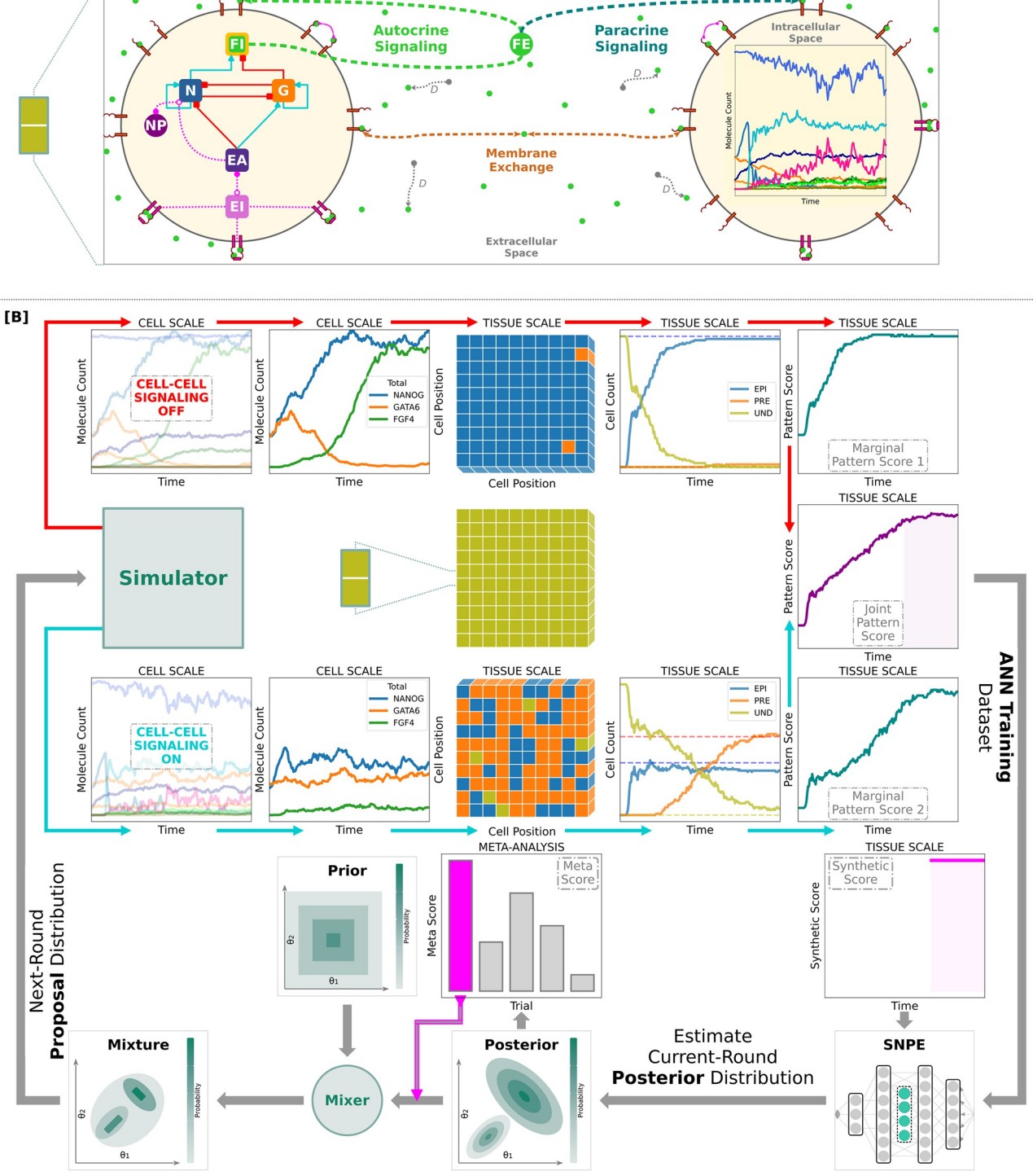

**Fig 1. Workflow summary: GRN motif, cell signaling model, and inference framework. [A]** The developing ICM is represented by a static spatial lattice of biochemical reaction volumes (cells) coupled via FGF4, which mimics a monolayer or 2D cellular culture. Each cell contains a core gene-regulatory network (GRN) featuring mutual repression between the genes *Nanog* (N) and *Gata6* (G), and their self-activation. Both N and G regulate the expression of internal FGF4 (FI). External FGF4 (FE) can either diffuse to a neighboring cell (paracrine signaling), bind to the membrane of the origin cell (autocrine signaling), or be exchanged between neighboring cell membranes. FGF receptors transmit the sensed FGF4 signal back to the core GRN by activating ERK (EI ↔ EA). **[B]** Pipeline of data generation and analysis. Key stages of parameter inference (columns 1 through 5). Initially, all cells display the undifferentiated (UND) fate (row 2 column 3). Rows 1 and 3 show simulations without and with FGF4 signaling activated, respectively. All generated stochastic trajectories are processed by the same steps: (I) resampling data onto a regular time grid and calculating relevant system observables

at cell scale (total NANOG, GATA6, and FGF4 levels); (II) determining the lineage for each cell at every time point (EPI, PRE, or UND); (III) summing up the corresponding total cell count for each fate at tissue scale; (IV) constructing the (joint) pattern score time series. The map between simulation parameters and resultant score time series is used for training a deep neural density estimator via the sequential neural posterior estimation (SNPE) algorithm (row 5 column 5), which directly estimates the parameter posterior distribution (row 5 column 3) conditioned on a target observation (row 4 column 5). Multiple posterior estimates are produced with the same training set, selecting the best learned distribution conditional on the target observation by analyzing a "meta score" distribution (row 4 col 3). This summary/meta score is calculated per posterior, and it relies on the maximum a posteriori (MAP) estimate of all the model parameters. In general, the next-round prior does not need to be the current-round posterior: it is plausible to obtain a well-informed next-round mixture distribution (row 5 col 1). Several iterations of the workflow are performed until the meta score surpasses an arbitrarily prescribed level. For additional information about this data-processing pipeline, please see Model parameter inference framework.

Simulation-Based Inference (SBI) frameworks have emerged as powerful strategies addressing several of these modeling and inferential challenges [89–92]. Particularly advantageously, SBI allows to learn multidimensional parameter sets that comply with a prescribed behavior formulated in terms of lower-dimensional utility functions, thus mitigating the limitations posed by the scarcity of detailed quantitative data.

Here we use the SBI framework [93–95] to establish a comprehensive parameter exploration workflow for our spatial-stochastic model of mouse ICM fate decisions. Notably, our approach does not depend directly on fitting quantitative experimental data. Instead, it primarily uses qualitative observations to reconstruct system behavior and infer parameter distributions complying with it in a holistic manner. At heart, our approach fuses a novel AI-based technique, specifically the Sequential Neural Posterior Estimation (SNPE) algorithm [86, 87], with concepts from classical inference and optimization strategies [96, 97]. The key steps of our approach are summarized by the following workflow:

(1). Construct an objective function that quantitatively represents the system dynamics as a time-varying "score", indicating the progression towards a desired or ideal state. For the ICM model studied here, this function traces the deviation from the desired cell-fate ratio (see "Constructing the pattern score (objective) function" for details).

(2). Simulate the model many times, each with a different parameter value vector sampled from a suitable prior distribution.

(3). Evaluate the objective function using the data obtained from these simulations.

(4). Train an artificial neural network (ANN) to learn the posterior distribution of the model parameters, effectively using the ANN as a surrogate for the simulator to approximate the mapping between parameter values and objective function scores.

(5). Define a target score time series that represents the ideal system behavior, using it as a synthetic observation.

(6). Estimate the maximum-a-posteriori (MAP) parameter value that aligns with the desired system behavior represented by the target score time series, using the posterior distribution predicted by the ANN conditioned on the given target score.

(7). Conduct multiple iterations starting from step (2) until the resulting score time series aligns with the target score time series.

For a more detailed description of these steps, see Fig 1 and "Model parameter inference framework". In a companion study [98], we compare this SNPE-guided methodology against a strategy inspired by a classical optimization algorithm, specifically simulated annealing; we find that the AI-based approach surpasses its classical counterpart in predictive performance, assuming equal allocation of computational resources. This superiority can be traced back to

the innate flexibility of the ANN and the absence of a native parameter interpolation procedure in simulated annealing.

**Reproduction of tissue-level features defines a score function.** We explored ICM differentiation guided by three pivotal characteristics which informed and constrained our parameter inference workflow.

The first key feature is the high reproducibility of the EPI and PRE cell-fate proportions in the blastocyst [3, 29, 38]. Despite varying reports on its precise ratio, here we adopted a typically reported value of 2 : 3 [99]. Note that minor deviations from this ratio are unlikely to significantly change the inferred posterior distributions or other findings of our model.

Secondly, blastocyst formation occurs within approximately 1.5–2 days of development, starting around E2.75 [2, 3, 30]. The EPI and PRE populations must reach and sustain the required fate proportions 8 to 12 hours before the end of the preimplantation period (around E4.75). This requirement serves as another optimization criterion for our model, regardless of the actual underlying cell-differentiation mechanism.

The third key feature is the role of spatial coupling via FGF4 in fate determination. In its absence, almost the entire ICM population assumes the EPI fate, which impedes the exit from naive cellular pluripotency [21, 73]. Correspondingly, only few cells adopt the PRE fate, a phenomenon attributable to intrinsic stochasticity at the gene expression level. This highlights the critical importance of local cell-cell signaling for correct pattern formation, and the inherent bias of the ICM population towards a naive pluripotent state.

All these key features together were incorporated into the design of the "score function" used in our SBI approach, in order to infer parameter distributions complying with them. The mathematical definition of the score function is described in Methods section "Constructing the pattern score (objective) function".

**Comparative analysis of parameter sets for complementary wild-type and mutant models via SBI.** Exploiting our SBI workflow, we successfully identified two distinct sets of model parameters that shed light on crucial aspects of the underlying biological problem. These two parameter sets correspond to two complementary models: one representing a wild-type system with functional cell-cell communication via FGF4/ERK, referred to as "Inferred-Theoretical Wild-Type" or ITWT, and another mimicking a mutant system without FGF/ERK signaling (more precisely, without cell-cell FGF4 signaling), referred to as "Reinferred-Theoretical Mutant" or RTM. The RTM is a direct adaptation from the ITWT, but with reinferred core GRN interaction parameters; see also Tables 1 and 2.

Fig 2 summarizes the key differences between the ITWT and RTM systems, contrasting the respective inferred parameter values, i.e. the maximum-a-posteriori-probability (MAP) estimates, for both the central intracellular GRN components (top row) and the extracellular signaling topology (bottom row). The most notable difference between the ITWT and RTM models is the reversal of the relationship between the self-activation thresholds of *Nanog* and *Gata6*. For the ITWT, the half-saturation threshold of *Nanog* self-activation (*Nanog*_NANOG) is lower/stronger than the half-saturation threshold of *Gata6* self-activation (*Gata6*_GATA6). However, both models exhibit a well-balanced mutual repression between *Nanog* and *Gata6*, albeit with varying interaction strengths. We present a complete overview of the full model parameter distributions inferred in this study, and a first approximation of the model parameter sensitivities, in S1 Appendix as well as S1 and S2 Figs.

Both models correctly recapitulate the final ratio (of cell counts) between the two emerging ICM lineages (EPI and PRE) of the fully-formed mouse blastocyst. However, the ITWT relies on a cell non-autonomous mechanism in which spatial coupling via FGF4 is essential for accurate and precise lineage specification. In contrast, the RTM relies on a cell autonomous mechanism, which is purely probabilistic. We explore this mechanistic difference and its

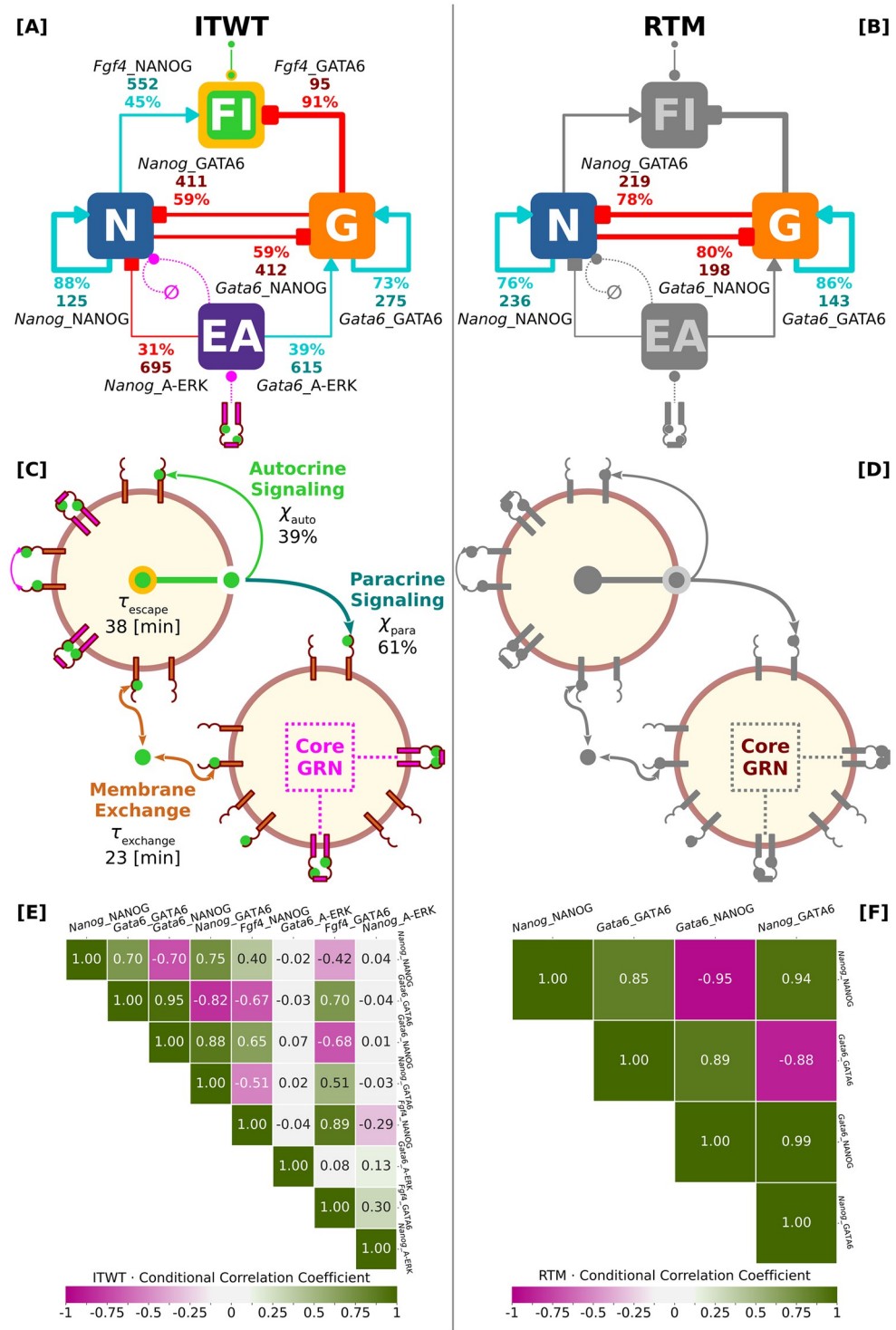

**Fig 2. Summary of inferred central GRN and signaling model parameters (ITWT versus RTM). [A-D]** Two parameter sets were learned for distinct systems capable of recapitulating the correct final ratio between the emerging ICM lineages (EPI and PRE). The wild-type-like system ITWT (inferred-theoretical wild-type—left column) with functional cell-cell signaling via FGF4 (green color), and the mutant-like system RTM (reinferred-theoretical mutant— right column) for which FGF4 signaling was inhibited. Red- and cyan-colored numbers show the inferred values (MAP estimates) of the relevant parameters defining the central intracellular GRN (top row) and the extracellular signaling topology (mid row) components. Red- and cyan-colored percentage values indicate relative interaction strengths calculated via the formula $\nu = 100(\omega - \theta)/\omega$. Here $\omega$ is the length of the respective parameter range used for

the inference scheme, and $\theta$ is the MAP estimate of the particular parameter. Cyan and red colors refer to gene activation and repression, respectively. **[E, F]** Conditional model parameter correlation matrices (ITWT versus RTM —bottom row). Pearson's correlation coefficients computed after conditioning posterior distributions on their own MAP estimates. Note high (absolute) linear correlations between *Nanog*_NANOG, *Gata6*_GATA6, *Gata6*_NANOG, and *Nanog*_GATA6, suggesting potential compensation mechanisms among parameter-value pairs.

consequences in more detail below (sections "Intercellular communication via FGF4 functionally improves ICM differentiation robustness by 10–20% compared to a purely-binomial baseline scenario" and "Robust cell-fate proportions are independent of cell-grid size"). In the following, however, we focus on the ITWT, as it incorporates the necessary FGF4 signaling confirmed by experiments.

**Posterior distributions approximated via SBI reveal correlations and potential compensatory mechanisms between model parameters.** To illustrate a key advantage of our AI-powered inference method, which extends beyond simply providing point and interval estimates of model parameters aligned with a postulated target behavior, here we exemplify the generative capabilities of an ANN trained via the SNPE algorithm. Specifically, we compute the correlation matrix of parameter interactions for the two inferred models, ITWT and RTM (see Fig 2E and 2F). The trained ANN serves as a direct surrogate for the posterior distribution consistent with the target behavior, and can be easily conditioned on the optimal model parameter set represented by its MAP estimate (see Fig 2A–2D).

This conditional posterior is particularly useful for exploring parameter space structures, as it can generate linear correlation coefficients between any pair of model parameters, thereby uncovering compensation mechanisms that preserve the desired system behavior. More importantly, since the trained ANN functions as an implicit surrogate for our spatial-stochastic simulator, this analysis is computationally efficient and feasible, unlike traditional parameter sensitivity or perturbation analyses that typically depend on brute-force techniques that require extensive additional simulations [89].

For brevity, we focus on the core gene regulatory network (GRN) interactions for both ITWT and RTM models, as depicted by the size of their respective conditional correlation matrices (Fig 2E and 2F). A comprehensive analysis of the entire parameter space structure is presented in our companion study [98]. Here, we highlight the most significant predictions derived from our approach using the estimated full model parameter posterior distributions. These predictions reveal potential compensatory mechanisms among parameter pairs that ensure the robustness and reproducibility of the target patterning behavior.

For the ITWT model (Fig 2E), we highlight the strong flexibility of the simulated biophysical system in response to perturbations of core GRN parameters, specifically *Nanog*_NANOG, *Gata6*_GATA6, *Gata6*_NANOG, and *Nanog*_GATA6. This flexibility is evidenced by their pairwise high absolute conditional correlation coefficients, which is notable given their highly nonlinear interactions. The trained ANN has successfully learned the complementary relationships among these parameters. For instance, weakening an auto-activation threshold can be counterbalanced by strengthening its associated mutual repression threshold to preserve overall system behavior, as shown in Fig 2E by their anti-correlated interactions. Furthermore, the trained ANN predicts the existence of linear hyperplanes where different parameter values can effectively compensate for each other, creating a large population of behavior-conserving models. This underscores the strong perturbation resilience and adaptability characteristic of real biological systems [95, 100, 101]. These resilience and adaptability traits emerge in the presence of not only model parameter variations but also intrinsic and extrinsic sources of biochemical noise.

The ITWT conditional correlation matrix also predicts a high interdependence of parameter value choices for the secondary core GRN parameters, particularly *Fgf4*_NANOG, *Fgf4*_GATA6, and *Nanog*_A-ERK, but excluding *Gata6*_A-ERK. The activation and repression thresholds for *Fgf4* must move in tandem, as they are positively correlated. However, these regulatory thresholds of *Fgf4* are correlated with opposite signs with the auto-activation thresholds of either *Nanog* or *Gata6* (compare first and second entries in columns *Fgf4*_NANOG and *Fgf4*_GATA6, respectively).

Note that while we purposefully included the repression of *Gata6* expression by active ERK protein, motivated by recent in vitro experiments [102], our correlation analysis predicts that this interaction may be redundant or unnecessary for achieving and replicating the target system behavior. Indeed, the true role of this interaction remains a topic of debate in several experimental and theoretical studies [9, 15, 38, 69]. This observation underscores the potential utility of our approach for discovering and validating topological structures of similar GRNs.

For the hypothetical RTM model (Fig 2F), similar observations can be made. Here, the linear coefficients are expectedly higher compared to the ITWT case, as the recapitulation of the target behavior relies solely on the proportional compensation of four parameter values. Since adjusting cell-cell signaling is not possible in the RTM model, retaining proper cell fate proportioning requires to entirely focus on the activation and repression thresholds. As in the ITWT model, the trained ANN predicts clear complementary linear relationships among primary core GRN parameter pairs.

## EPI and PRE fates emerge robustly at tissue scale in spite of high single-cell variability

Our simulations were initiated with all cells in an undifferentiated (UND) state with a balanced distribution of cellular resources. At cell scale, the first part of a typical stochastic trajectory reveals the dynamic interplay between NANOG and GATA6 proteins, with FGF4 protein expression being adjusted in response to the levels of these two pivotal regulators. After about 12 hours, on average, the initial symmetry between NANOG and GATA6 is broken, as their proteins embark on divergent expression paths; this is exemplified for the case of high GATA6 and low NANOG final expression in Fig 3A and 3B. As time progresses, the individual cells predominantly commit to a PRE or an EPI fate, accompanied by a decrease or increase in FGF4 expression, respectively. This differentiation is clearly depicted at the tissue scale, with cells categorically aligning into one of three fates: UND, EPI, or PRE (Fig 3B).

The variability in the expression profiles of NANOG, GATA6, and FGF4 at the cell scale is significant, as shown by the large standard deviation in molecular counts across different cells and simulations (Fig 3C). However, this cell-scale variability does not translate into high variability of the cell fate ratio at the tissue scale, which instead exhibits remarkable robustness, with a significantly lower standard deviation (Fig 3D). This observation alone underscores the system's ability to integrate and manage cellular variability, ensuring consistent and reliable outcomes in the differentiation process across the tissue.

## EPI cells precede and are necessary for specification of PRE cells

Transitioning from this foundation of robust differentiation, the model predicts an early and critical onset of the EPI lineage, occurring around 2 hours into the simulated trajectories, with the PRE lineage emerging only about 12 hours later. The emergence of the first EPI cells is tightly clustered within a narrow time frame between 2 and 3 hours of development (gray region in Fig 4A); this temporal behavior is consistent across a wide range of initial conditions (detailed in "Computational experiments"). Following this timely commitment, the EPI

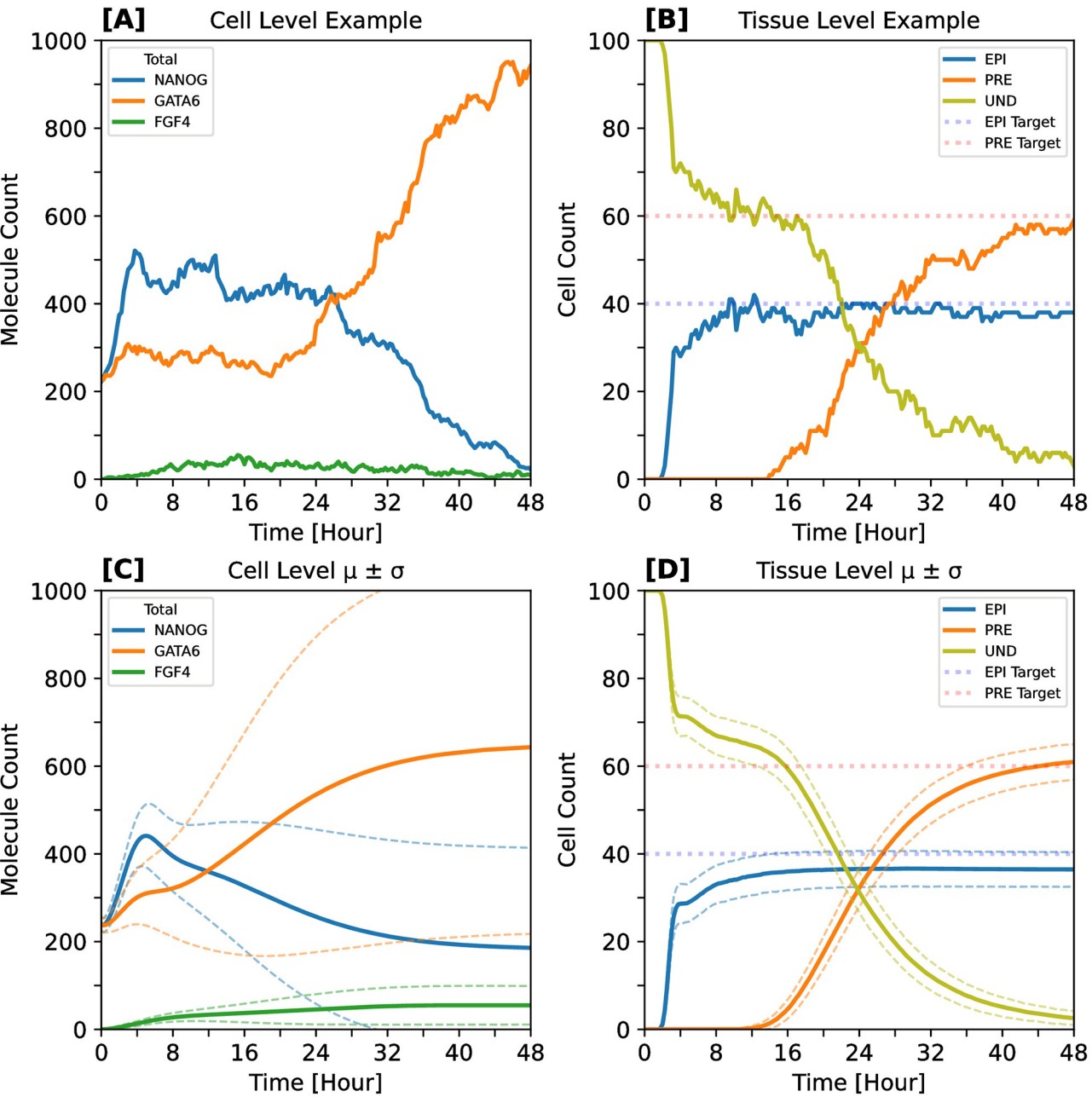

**Fig 3. Stochastic trajectories at cell and tissue scale.** The inferred-theoretical wild-type (ITWT) system shows excellent agreement with various experimentally observed characteristics of ICM cell specification. All simulations start with all cells in the undifferentiated (UND) state, having well-balanced cellular resources, before committing exclusively to EPI or PRE fates. **[A]** Example cell-level stochastic trajectory, randomly taken from a 100-cell (10×10 grid) simulation. Within the first 24 h, the cell remains in the UND fate while the FGF4 protein level increases. At around 24 h, a symmetry-breaking event occurs, as the NANOG and GATA6 protein levels take divergent expression paths. Within the last 24 h, the cell acquires the PRE fate while FGF4 levels decrease again. **[B]** Example tissue-level stochastic trajectory of cell fate counts in a 10×10 cell-grid system. Each cell is categorized into one of three possible fates: UND, EPI, or PRE. With progressing time, UND cells reduce in number while EPI and PRE cell counts settle close to prescribed constant target levels (dotted lines); see Methods section "Computational experiments" for details of cell-fate classification. **[C]** Typical cell-level behavior of NANOG (blue), GATA6 (orange), and FGF4 (green) total protein levels. **[D]** Typical tissue-level behavior of EPI (blue), PRE (orange), and UND (olive) cell-fate counts. Solid lines represent means, dashed lines represent standard deviations around means. Statistics are computed from a batch of 1000 simulations of a 100-cell tissue system.

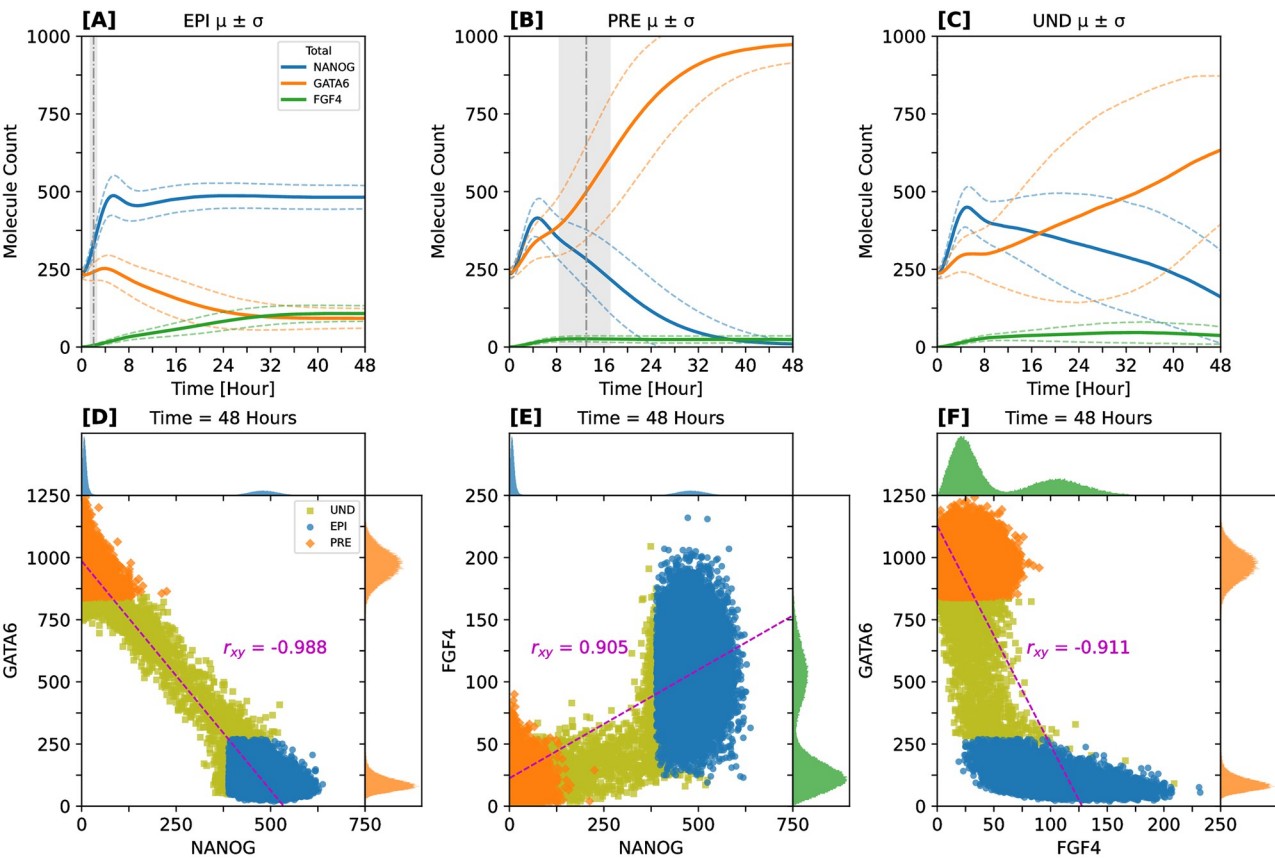

**Fig 4. EPI cells precede and are necessary for specification of PRE cells. Expression profiles reveal strong linear correlations among key proteins.** Upper row shows typical cell-level protein count time evolution for each cell-fate category. Cell fates were identified at 48 h. For each fate, cell-level dynamics were traced from last to initial simulation data point. **[A-C]** Solid lines represent mean behavior, dashed lines represent standard deviations around means. Gray regions accentuate time intervals of first differentiated cell appearance for the given fate (vertical dash-dotted line = mean time). Statistics are computed from a batch of 1000 simulations. Only the most relevant proteins are shown: FGF4 (green), NANOG (blue), GATA6 (orange). **[D-F]** Pairwise relationships among the most important proteins at 48 h. Inner panels show different colors representing the protein-pair relationship for the respective cell fate: UND (olive), EPI (blue), PRE (orange). Outer panels show protein count histograms. Data points come from a batch of 1000 simulations. $r_{xy}$ = Pearson's product-moment correlation coefficient. For details of cell-lineage classification, see Methods section "Computational experiments".

population expands rapidly, attaining, on average, 75% of its target proportion within the initial 4 hours (see Fig 3D). This leads to elevated FGF4 levels among the newly specified EPI cells, enabling the distribution of FGF4 across the ICM. Only then PRE cells begin to appear in significant numbers within a broader time window, ranging from 8 to 17 hours (refer to the gray region of Fig 4B), implying an approximate 7-hour delay between the emergence of the first EPI and PRE cells, in line with recent experiments [30]. Subsequently, the PRE cell population gradually increases, reaching its target proportion at around the 40-hour mark (Fig 3D).

Nascent PRE cells are coordinated by EPI cells via differential expression of *Fgf4*, and the expression profile of *Gata6* is a principal indicator of the onset of PRE cells. This coordination is controlled by nuances in fate-specific FGF4 distributions. Such nuances not only control the emergence of PRE lineage, but they are also key for EPI- and PRE-fate maintenance [9, 21]. Tight control of *Nanog* expression in EPI cells is a requirement for escaping naive pluripotency during the implantation stage [103–105].

Note that in our model the observed precedence of EPI emergence over PRE cells arises naturally as a predictive result, without any explicit incorporation into the modeling or inference

procedures. The mechanism driving the delayed emergence of the opposing cell fates can be summarized as follows: first, stochastic self-activation of *Nanog* triggers the EPI fate specification program in a subset of the ICM cells. This, in turn, promotes the progressive differentiation of other unspecified cells into the PRE fate when they sense FGF4, which is only released by cells where *Nanog* reached substantial expression levels. A crucial precondition of this mechanism is the lower self-activation threshold of *Nanog* compared to *Gata6*. Despite the inherent stochasticity of the differentiation process at the single-cell level, coordination via FGF4 makes it appear deterministic at the tissue level.

## Expression profiles reveal strong linear correlations among key regulatory proteins

Our simulations show that the copy number distributions of the three key proteins NANOG, GATA6, and FGF4 are clearly bimodal by the 48-hour mark, as depicted in Fig 4D–4F. At the beginning of every simulation, we use a well-defined initial condition distribution (ICD) which restricts the protein and mRNA expression levels to a region where all the cells start with the undifferentiated (UND) fate. This guarantees symmetric splitting of initial resources on average and prevents any systematic fate bias at the simulation start. Despite the innate randomness of the ICDs employed for simulating, early variability does not have adverse effects in the final lineage proportions. Instead, contrasting expression profiles slowly emerge and are clearly visible by the last simulation time point (48 hours).

A notable observation from our simulation data are strong linear correlations among these key regulatory proteins, which emerge in spite of the nonlinear regulatory interactions between them. Specifically, we identify a pronounced negative linear correlation between NANOG and GATA6, as well as between GATA6 and FGF4 (refer to Fig 4D and 4E). Conversely, a strong positive linear correlation is observed between NANOG and FGF4 (see Fig 4F).

Affirming the validity of our simulation approach, these robust linear relationships mirror experimental findings: Fig 5D in [65] shows similar trends among *Nanog*, *Gata6*, and *Fgf4* mRNA levels at ∼ 64-cell stage, using sc-qPCR data; Fig 2B and Fig 3B in [66] show correlated mRNA levels of EPI-PRE markers at ∼ E4.5 and the anticorrelation between NANOG-GATA6 protein levels at ∼ E3.5, respectively, employing sc-qPCR data; Fig 4B in [68] shows negative correlation between *Gata6* and *Sox2* (proxy for *Nanog*) mRNA levels at ∼ E4.5, utilizing scRNA-seq data; Fig 2B in [9] shows negative correlation between GATA4-mCherry (proxy for GATA6) protein and *Fgf4* mRNA levels at multiple induction times, using qHCR imaging. We remark that our simulation data go beyond these available experimental measurements, because we provide a more granular and completely quantitative perspective on the correlations among these key regulatory elements.

## Intercellular communication via FGF4 functionally improves ICM differentiation robustness by 10–20% compared to a purely-binomial baseline scenario

In order to investigate the robustness to noise in ICM specification, we assessed whether and to which extent tissue-level coupling via FGF4 is capable of reducing variability in the acquired cell fates. To this end we compared the variability observed in our Inferred-Theoretical Wild-Type (ITWT) model simulations to an entirely cell-autonomous fate decision-making scenario. In such hypothetical "Purely Binomial" (PB) scenario, the cell lineage distribution is supposed to follow a binomial pattern, as each cell's fate, either EPI or PRE, is determined independently of others. The comparison was carried out by analyzing the coefficient of

variation (CV) of 48-hour fate proportions across 13 different system sizes ($\eta$ = [5, 10, 15, 25, 35, 50, 65, 75, 85, 100, 150, 225, 400] cells) for two cases.

For the first case, we calculated the coefficient-of-variation ($CV_1$) as the ratio of sample standard deviation to sample mean. For the second case, the hypothetical PB scenario, the corresponding measure ($CV_0$) is simply the coefficient of variation of the binomial distribution, with standard deviation being a function of mean fate numbers and total cell count. Interestingly, for the EPI and PRE fates (excluding the UND category) $CV_1$ is consistently lower than $CV_0$, which is indicative of noise reduction due to FGF4 signaling (see Fig 5A). The ratio $CV_0/CV_1$ suggests that the ITWT model outperforms the PB model by 10–20%, implying fewer incorrectly specified cells (inset of Fig 5A).

To further validate the role of cell-cell communication in enhancing patterning robustness, we also compared the ITWT model to the Reinferred-Theoretical Mutant (RTM). The RTM lacks cell-cell signaling (see Fig 2 and S1 Fig, plus Methods section), reproducing the prescribed cell-fate ratio (on average) with a purely cell-autonomous patterning mechanism. We asked whether binomial noise emerges naturally in this system. Indeed, we found that the CV of the RTM model is comparable to that of a PB model (Fig 5C). Both systems adhere to the same power law, with negligible differences across system sizes (inset of Fig 5C).

In conclusion, our findings demonstrate that cell-to-cell communication via FGF4 diffusion, encoding local environmental variations, enhances ICM fate differentiation robustness by approximately 10–20% compared to a purely cell-autonomous scenario. This finding corroborates the notion that ICM differentiation is a tissue-level process, where tissue-scale signaling feedback via FGF4 plays a functional role in mitigating cell-fate decision noise. At the same time, it is in line with previous studies highlighting the benefit of spatial coupling for noise reduction in developing tissues [46, 47, 54, 55, 106–108].

## Robust cell-fate proportions are independent of cell-grid size

Recent experiments suggest that robust control in the EPI to PRE lineage ratio does not depend on the absolute size of these populations [29, 99]. Resilience of the mouse embryo to variations in ICM size, as reflected by alterations in total cell number, was found both in vivo and in silico [1, 8]. Nonetheless, there remains a debate on whether a critical embryo size is essential for proper blastocyst lineage segregation [7, 13, 22, 109]. To assess the impact of absolute tissue size (cell number) on ICM specification, we analyzed cell-fate proportions and associated noise levels across various tissue sizes, hypothesizing that smaller cell numbers might correlate with increased noise in fate decisions.

We conducted 1000 simulations for each of 13 distinct cell grid sizes, ranging from 5 to 400 cells in total, for both the ITWT and the RTM models. We find that noise intensity scales with system size with a power law in both models, as illustrated in Fig 5A and 5C. This indicates that noise diminishes predictably as system size increases.

Moreover, our simulations reveal a universal mean value ($\mu$) for cell-fate proportions, consistent across all different cell grid sizes for both the ITWT and RTM. Despite this, the two models show, respectively, unique characteristics in commitment times, standard-deviation magnitudes, and independence of fate choice among cells (see Fig 5B and 5D).

The commitment time discrepancies between the ITWT and RTM models can be attributed to their distinct mechanisms. The ITWT augments probabilistic differentiation with tight control via FGF4 signaling, which makes it more resilient against perturbations. This mechanism requires initial random emergence of a portion of the EPI population, which subsequently coordinates other undifferentiated cells towards specific fates based on local neighborhood information, thus globally regulating EPI-PRE proportions. Early commitment to the EPI fate

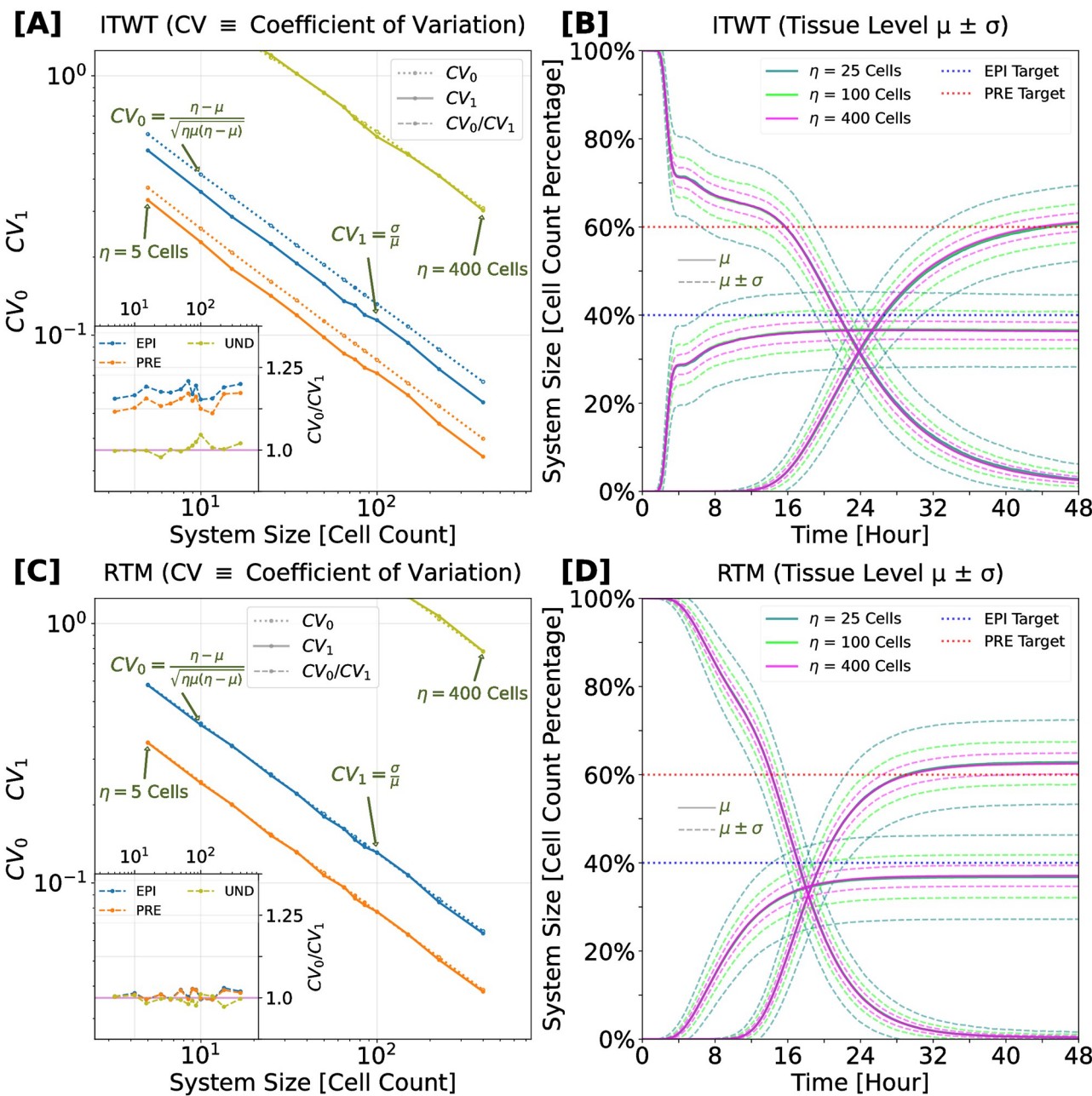

**Fig 5. Noise at tissue level: intercellular communication improves ICM differentiation robustness by 10–20% compared to purely-binomial baseline scenario.** [**A**] Comparison between inferred-theoretical wild-type (ITWT ∼ $CV_1$) and purely-binomial (PB ∼ $CV_0$) systems. The main plot shows the coefficient of variation (CV) as a function of the system size for each cell fate (colors as in Fig 4). The inset shows the ratio between $CV_0$ and $CV_1$, highlighting systematically lower fate specification error in the ITWT compared to the PB system. [**C**] Comparison between reinferred-theoretical mutant (RTM ∼ $CV_1$) and purely-binomial (PB ∼ $CV_0$) systems. Colors and symbols as in [**A**]. The inset plot shows that fate specification errors in the spatially uncoupled RTM system are of the same magnitude as in the PB scenario. Axes use logarithmic (base 10) scale. For each system size, the data point is computed from a batch of 1000 simulations. [**B, D**] Typical time evolution of normalized cell fate counts for three example system sizes ($\eta \in [25, 100, 400]$ cells). Solid lines represent means, dashed lines represent standard deviations around means.

and subsequent emission of FGF4 is crucial to this process. In contrast, the RTM relies purely on stochastic differentiation, and is more sensitive to perturbations. This system lacks regulation beyond the inherent genetic program at the cellular level and does not integrate tissue-neighborhood information, with fate commitment timing primarily dictated by target protein levels for EPI and PRE markers.

The computationally predicted scaling rule in Fig 5 could potentially be tested in vitro using current experimental techniques. In a wild-type-like system, one could measure the mean and standard deviation of specified cell numbers as a function of the overall population size at predefined time points. This could also be done in a mutant-like system with cell-autonomous EPI-PRE differentiation (lacking cell-cell signaling), however here the key challenge would be the creation of such system in the first place. This could be facilitated by recent advances in optogenetic expression control for altering the signaling network in a targeted fashion [110, 111]. For both systems, the predicted scaling could be assessed by artificially splitting the original cellular population into subsets with distinct cell numbers, e.g. by creating grids with varying numbers of communicating cells (for the WT system), or uniformly randomly sampled cellular neighborhoods of varying size (for the mutant system). This experimental strategy would mimic modern bootstrapping techniques [112].

We emphasize that, unlike the purely cell-autonomous case where binomial noise is expected, the predicted scaling in the spatially coupled case is non-trivial. Intercellular communication in the ICM is a nonlinear phenomenon influenced not only by fluctuating copy numbers of key intracellular molecular players but also by response delays arising from the production and transport of signaling agents. This necessitates integration of stochastically fluctuating neighboring information.

In sum, our findings suggest no critical cell number for accurate ICM fate specification; however, its precision increases with the (square root of the) cell number, while spatial coupling via FGF4 can reduce the noise magnitude by $\sim 20\%$ compared to a purely cell-autonomous mechanism.

## Autocrine- and paracrine-signaling modes play reciprocal roles in robust cell-cell communication

One key characteristic of the mouse blastocyst is the overwhelming dominance of EPI cell fates when FGF4 production is inhibited or related loss-of-function mutations are applied. In such cases, almost all cells commit to the EPI fate by the time of implantation, as they are unable to exit naive pluripotency due to the absence of mechanisms controlling precise *Nanog* expression, leading to adverse developmental outcomes [21, 73, 105, 113, 114].

In agreement with this (and as demanded by the imposed score function), when FGF4 signaling is impeded in our simulations, cells initially co-express EPI and PRE fate-specific markers, but eventually only a small subset adopts the PRE fate [9, 21]. This leads to NANOG upregulation and commitment to the EPI fate in the majority of ICM cells [8, 30].

To dissect the roles of different communication modes in ICM specification, we modified the ITWT model to interrupt specific components of the signaling pathway. This model incorporates three distinct communication modes between neighboring cells: autocrine, paracrine, and intermembrane ligand-exchange (akin to juxtacrine) signaling via FGF4. Autocrine communication occurs when a cell secretes signaling molecules that bind to receptors on its own membrane, thereby regulating its own activity. Paracrine communication, in contrast, involves the release of signaling molecules that act on nearby cells within the local environment, facilitating coordinated responses among neighboring cells. Intermembrane ligand-exchange

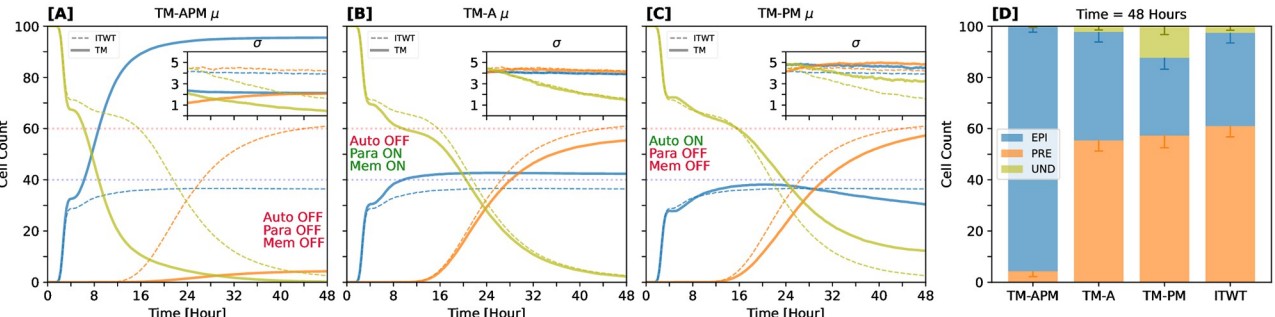

**Fig 6. Autocrine- and paracrine-signaling modes play reciprocal roles in robust cell-cell communication.** The figure shows the effects of perturbing autocrine- and paracrine-signaling modes in theoretical mutant (TM) systems. **[A]** The TM-APM (inhibition of autocrine, paracrine, and membrane-exchange signaling) represents the loss-of-function phenotype of experimental FGF/ERK pathway mutants. **[B, C]** The complementary TM-A and TM-PM portray the importance of individual feedback modes (paracrine plus membrane exchange and autocrine signaling only, respectively; see also inset text). Solid lines represent mean and standard deviation values ($\mu$ and $\sigma$) for a given TM; dashed lines represent $\mu$ and $\sigma$ for the inferred-theoretical wild-type (ITWT). **[D]** Cell-lineage allocation at the 48 h time point. All TMs are compared against the (ITWT). For all compared systems, statistics were calculated from 1000 independent simulation runs. Colors represent different cell fates: UND (olive); EPI (blue); PRE (orange).

occurs between adjacent cells when signaling molecules unbinding from the membrane of one cell are caught by receptors on the membrane of the neighboring cell.

In this way, we created three "theoretical mutants" implementing the following signaling scenarios: complete absence of FGF4 (TM-APM), lack of autocrine signaling (TM-A), and absence of paracrine signaling and membrane-to-membrane exchange (TM-PM). Each modification affects cell fate determination differently, as shown in Fig 6A–6C.

As expected, the TM-APM model exhibits a strong bias towards the EPI fate (Fig 6A and 6D). Initially, its dynamics parallel those of the ITWT system, but as the simulation progresses, the EPI fate predominates, with only a minor fraction of cells adopting the PRE fate.

The TM-A model displays a decreased accuracy of the ICM specification mechanism due to the absence of self-regulation, though the overall precision of the system remains unaffected (Fig 6B and 6D). Adjustments in other signaling components could potentially correct for this, but these also would likely reduce the system's ability to buffer against dynamic signaling perturbations.

In the TM-PM model, the critical role of paracrine communication and, to a lesser extent, FGF4 membrane exchange becomes evident (Fig 6C and 6D). Eliminating these communication modes disrupts the maintenance of target cell-lineage ratios, even when autocrine signaling is preserved. During initial simulation phases, differentiation seems normal, but over time, a significant fraction of cells remains undifferentiated, and the EPI lineage fails to sustain its population. Exchanging FGF4 with neighboring cells therefore is crucial for correct cell-fate specification, once again underpinning the importance of its tissue-scale coordination.

Although cells can only indirectly distinguish between autocrine and paracrine molecules (via receptor specificity-tuning or localization, and spatio-temporal patterns), it is plausible to quantify their particular roles played in proper EPI-PRE specification through the adaptation of recent bioengineering and programmable synthetic biology tools that emulate cell-cell communication [111, 115, 116].

Our findings predict that, as shown in Fig 6B and 6C, the absence of autocrine mode affects accuracy of cell-type proportioning, decreasing it by approximately 20% (while precision or noise remains unchanged). In contrast, the absence of the paracrine mode impacts both accuracy and precision, deteriorating homeostatic capabilities by approximately 30%. These

magnitudes consider the collective distortions across all target cell-lineage populations (EPI--PRE-UND) at the final simulation time point.

In summary, both autocrine and paracrine signaling are integral to ICM differentiation and maintenance. Autocrine signaling ensures the accuracy of fate specification, while paracrine signaling, along with membrane exchange, maintains lineage proportions, enhances precision, and promotes cellular homeostasis.

## ICM cells produce similar local and global neighborhood features: lineage ratios are preserved at both scales

We next asked whether the tissue-level spatial coupling via FGF4 leads to specific signatures in the emerging spatial distribution of cell fates.

A central challenge in developmental biology is the precise characterization of spatial patterns, such as the "salt-and-pepper" arrangement reported for the ICM, which has often been described informally in the literature [11, 16, 38, 117, 118]. The term typically implies a random distribution, yet randomness in a mathematical context can take various forms. Recent studies have endeavored to rigorously define this pattern using experimental and theoretical approaches [10, 25, 35, 85, 119]. Here we understand the "salt-and-pepper" pattern as an archetype in which each individual cell-fate decision is independent of the cell fates of its neighbors, which implies a multinomial distribution of cell fates in every tissue neighborhood.

The dynamically growing ICM is also shaped by cellular division and intercellular forces, which can lead to local fate clustering and compositional variability, as reported in prior studies [11, 35, 37]. In such scenario, a multinomial or "salt-and-pepper" distribution is not expected in the first place. However, here our static cell arrangement isolates the problem from these factors, and allows for an analysis that focuses solely on the influence of cell-cell communication on the spatial distribution of cell fates. We therefore asked to which extent the spatial cell distribution in our simulated ICM system agrees with or deviates from a multinomial baseline.

To this end, we first determined the neighborhood composition in our Inferred-Theoretical Wild-Type (ITWT) model, focusing on the three cell fate categories: EPI, PRE, and UND (Fig 7A–7C and 7E–7G). For each cell within a specific category, we included neighbors up to a predetermined degree (first or second-degree neighbors, as seen in Fig 7A–7C and 7E–7G; see Fig 7D and "Model at tissue scale" for the details of neighborhood stratification). We then compared the resulting cell-fate arrangements to the distributions resulting from artificially generated systems, in which the cell fates were sampled from a multinomial distribution with the same proportions as extracted from the simulated data. This comparison was carried out for all simulated times.

A subtle discrepancy, especially at the first-degree neighborhood level, emerged between the non-cell-autonomous system model ($\rho_1$) and the cell-autonomous, multinomial model ($\rho_0$) after 24 hours of simulation (inset plots of Fig 7A–7C). This difference, about ±3% in average neighborhood composition, is significant when considering their sampling distributions (standard errors are around ±0.5%). However, when comparing to the variation between simulations (with a ±10% standard deviation), this discrepancy becomes less significant.

Furthermore, we analyzed the FGF4 distribution across five independent neighborhood degrees ([0, 1, 2, 3, 4]; Fig 7H). After 48 hours, FGF4 predominantly remained concentrated near its source (EPI cells) but also spread to more distant neighboring cells at a notably lower molecular count. This finding aligns with experiments reporting that, in artificial systems, FGF4 signaling proteins stabilize around their source cells at a single cell-distance length scale [9]. It suggests that diffusive coupling balances the overall FGF4 level across the system,

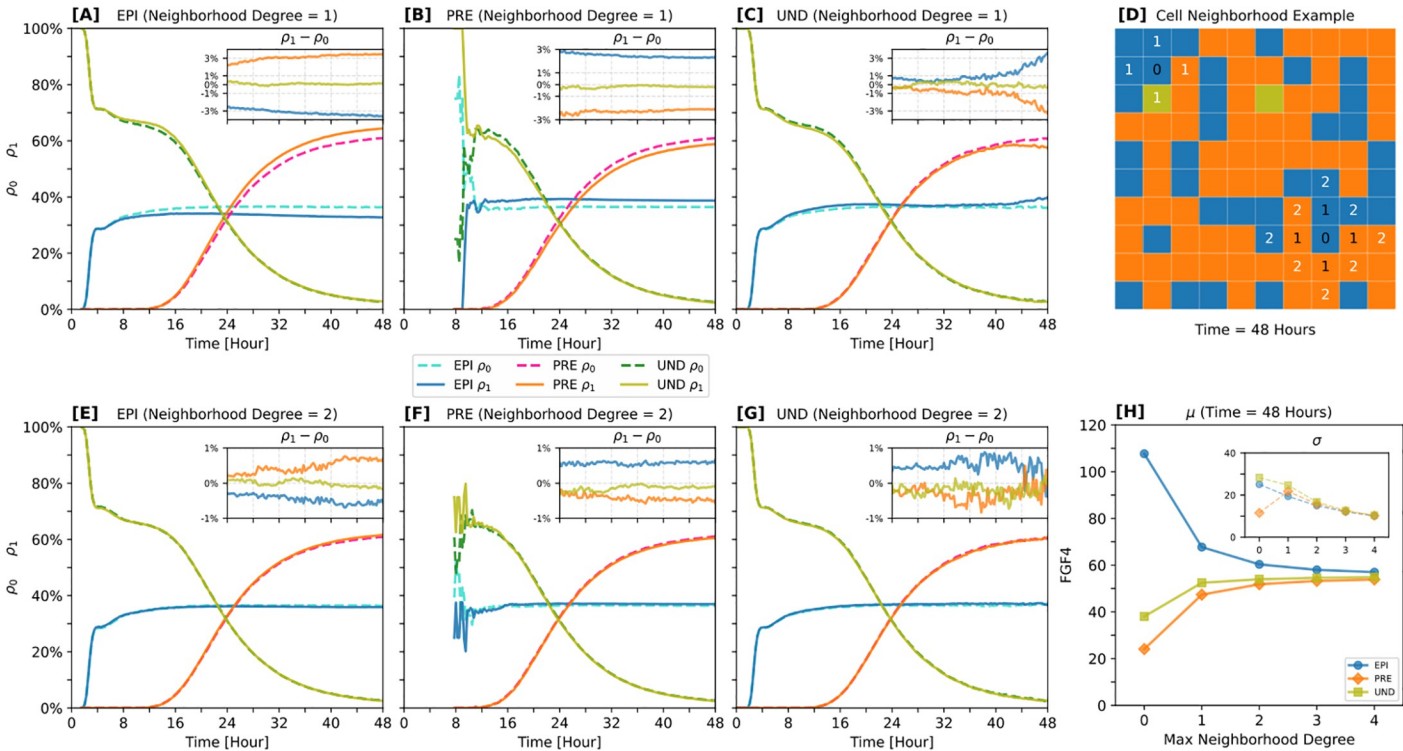

**Fig 7. Exploring self-similarity and spatial dynamics of simulated ICM neighborhoods. [A-C, E-G]** Temporal evolution of spatial correlations among acquired cell fates. Panels [A-C] (upper row) and [E-G] (lower row) show cell-neighborhood degrees 1 and 2, respectively (see also panel [D]). Here, $\rho_1$ and $\rho_0$ represent mean cell-neighborhood composition values split by fate: $\rho_1$ is directly computed from the spatio-temporal distribution of cell fates in inferred-theoretical wild-type (ITWT) simulations; in contrast, $\rho_0$ represents a multinomial baseline model assuming that cell fates emerge in a spatially uncoupled fashion from independent trials leading to the acquisition of one of three cell fates with fixed occurrence probability. The latter is estimated by averaging cell fate proportions at every time point across simulations, discarding any spatial information. **[D]** Example of a cell neighborhood at the 48-h time point. Numbers indicate the neighborhood degrees (1 and 2) relative to the reference cell (0). For details on the neighborhood stratification methodology, see Model at tissue scale. **[H]** Mean and standard deviation of FGF4 protein count as functions of maximum cell-neighborhood degree. FGF4 levels quickly become uncorrelated beyond the nearest-neighbor cell distance. Colors represent different cell fates. $\rho_0$: EPI (turquoise); PRE (deep pink); UND (forest green). $\rho_1$: EPI (blue); PRE (orange); UND (olive).

essentially acting as a quorum signal that reflects the proportion of FGF4-producing cells at the tissue scale.

Our findings indicate that both at the local and global scales, the spatial distribution of cell fates is similar, irrespective of whether cell differentiation is coordinated at the tissue level, as in our ITWT system, or purely cell-autonomous. This observation seems counter-intuitive, given the necessity of cell-cell signaling for proper lineage establishment. However, this may be part of a strategy to withstand strong perturbations (like drastic changes in cell population) for which cell fate proportions must be maintained locally but coordinated globally, such that neighborhood characteristics are preserved at both scales. This results in a seemingly irregular pattern that nevertheless preserves cell-fate ratios in a spatially homogeneous fashion.

## Increased variability in initial conditions enhances developmental accuracy while sustaining its precision

We next assessed how variability in initial condition distributions (ICDs) affects the accuracy and precision of cell-fate specification in our ITWT model; here, "accuracy" is defined as the closeness of the average simulated EPI-PRE proportions to the target ratios, and "precision" refers to the variability of these proportions among simulation ensembles.

Our approach was guided by two aspects: firstly, the broad inferred posterior distributions of the parameters governing the ICDs, and their moderate sensitivity to value changes (S2D and S2H Fig); secondly, similar assessments that were carried out in previous models of mouse-blastocyst, where mainly the ICD variance was modulated [15, 22]. Taking this into account, we generated 1000 stochastic trajectories for 10 distinct ICDs, modifying their variance from 0% to 200% compared with the baseline value (see Fig 8A for examples); the details of ICD modulation are described in Computational experiments. We analyzed both the mean cell count (accuracy) and the corresponding standard deviation (precision) for each cell fate (Fig 8C and 8D).

Remarkably, increasing the variance of the ICD can, in some cases, positively influence cell-fate specification in the ITWT system. For example, in the 100% initial condition perturbation (ICP) scenario, the accuracy of EPI/PRE specification improves notably compared to the baseline (contrast dashed to solid lines in Fig 8C), while its precision remains unaffected (Fig 8D and inset).

Between the 25% and 75% ICPs we observe a systematic reduction of the PRE populations in favor of the EPI populations (Fig 8B). This can be attributed to the fact that with increasing ICP strength a larger subset of cells is initially biased towards the EPI fate. This trend is inverted as we proceed to stronger ICPs (150% and 200%), since now the initially available *Nanog* abundance quickly induces FGF4 production, which promotes the PRE fate. Notably, while different ICPs thus alter the cell-specification accuracy, the corresponding precision remains similar for all levels of ICP strength (error bars in Fig 8B and inset plot of Fig 8D).

Our findings indicate that when the ICD is perturbed within normal ranges expected from full protein induction, cell-specification accuracy can be improved without compromising its precision. However, if the ICD is perturbed beyond this, the excess initial resources negatively affect the accuracy, while precision remains unchanged. These observations align with various previous studies which highlighted that stochasticity can play a constructive role in biological systems [79, 81, 82, 120–127].

To test how initial-condition variability affects developmental accuracy and precision in vitro (or even ex vivo), one could utilize photosensitive reagents to control mRNA and protein activity in living cells (i.e., optogenetics) [110, 126]. This approach would allow systematic control of cellular initial conditions, for example, via overexpression of competing molecular species.

## Cell-fate assignment remains robust when less than 25% of cells start with perturbed initial conditions

Having established that moderate increases in the variability of initial condition distributions (ICDs) can be beneficial for robust cell-fate specification, we now turn our attention to understanding the limits of this robustness by examining the system's response to different formats of ICD perturbations. In these tests, while maintaining a constant tissue size of 100 cells, a varying number of cells (ranging from 1 to 100) are randomly selected for ICD modifications.

The first perturbation scheme linearly modifies both NANOG- and GATA6-related resources in tandem, with the new mean ICD values ranging from 0% to 200% of the typical initial resources (Fig 9A–9D). For clarity, we have labeled these scenarios based on their deviation from the standard ICD, such as -100%, 0% (reference), and 100%.

The second perturbation scheme involves a negative linear correlation between NANOG- and GATA6-related resources, creating scenarios where one of these resources is initially dominant (Fig 9E–9H). The range of adjustments spans from 200% NANOG and 0% GATA6 to 0% NANOG and 200% GATA6, again compared to the unperturbed initial conditions.

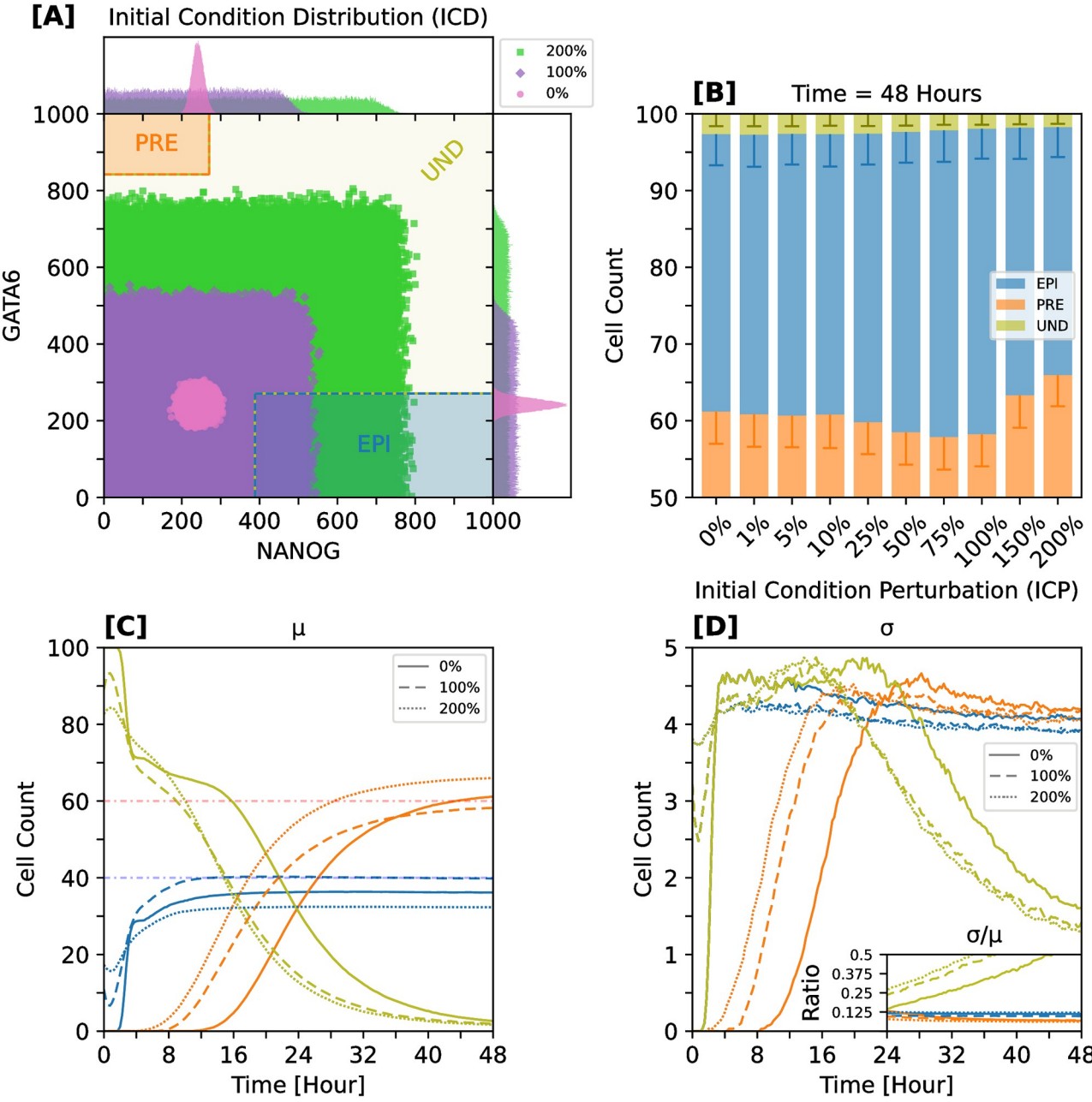

**Fig 8. Initial-condition variations enhance patterning accuracy and simultaneously sustain its precision.** Robustness to initial-condition perturbations (ICPs): uniform perturbation of protein (NANOG, GATA6) and mRNA (*Nanog*, *Gata6*) initial resources. **[A]** Example initial-condition distributions (ICDs) with increasing variability (see legend) for NANOG and GATA6 proteins. Interior scatter plots show sampling intervals; exterior plots show sampling distributions. The 0% distribution defines the unperturbed baseline case. For each ICD, data points come from 1000 independent simulations. **[B]** Mean cell-fate composition (bars) with standard deviation (error bars) for all tested variability intensities. Bar height ranges from 50 to 100 cells. **[C]** Time trajectories of mean cell-fate counts for selected ICD variability intensities. Dotted lines specify target proportions for EPI and PRE fates, respectively. **[D]** Time trajectories of cell-fate count standard deviations for selected perturbation intensities. Inset plot displays temporal coefficient of variation ($CV = \sigma/\mu$) for the last 24 h. Colors represent different cell fates. Statistics were calculated from batches of 1000 independent simulations per ICD.

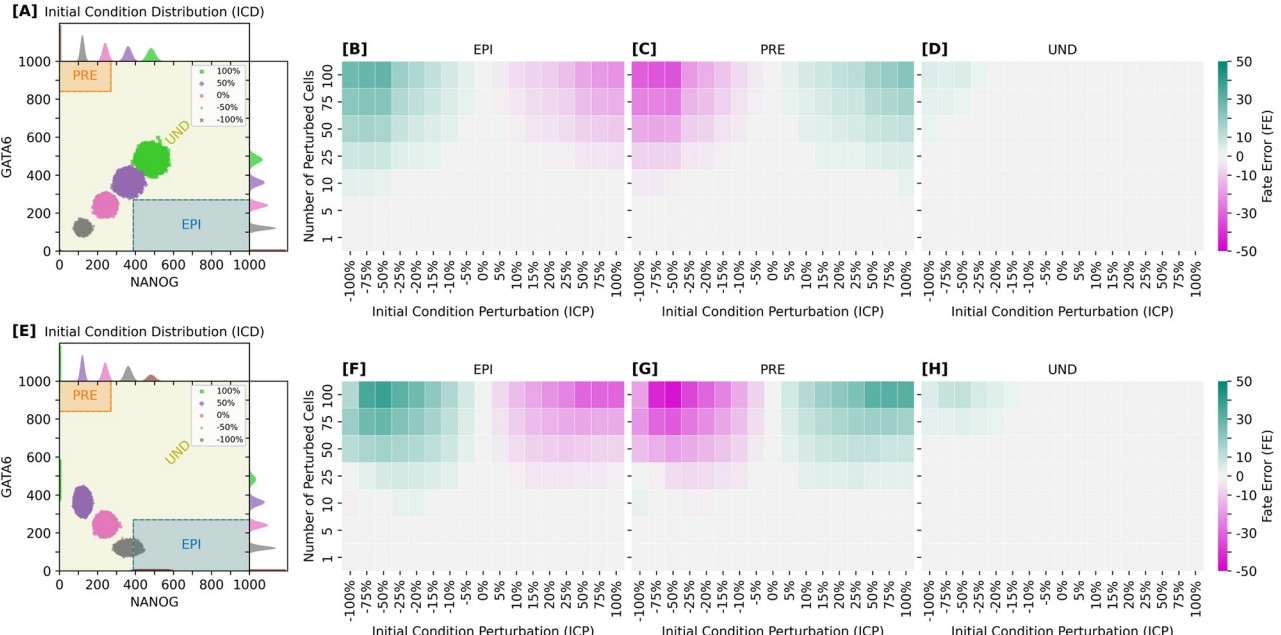

**Fig 9. Resilience to perturbations is mainly determined by number of affected cells: Patterning remains robust when less than 25% of cell population is perturbed.** Robustness to linear initial condition perturbations (ICPs) of protein and mRNA initial resources (NANOG-GATA6 and *Nanog-Gata6*). Upper row [A-D] shows the effect of equally perturbing resources of both genes. Lower row [E-H] shows adversarial perturbation of the resources, i.e. linearly progressing perturbation magnitude from 200% NANOG and 0% GATA6 (—100% ICP scenario) to 0% NANOG and 200% GATA6 (100% ICP scenario). [**A, E**] Initial condition distributions (ICDs) for NANOG and GATA6 proteins. Interior (scatter) plots show sampling intervals. Exterior plots show sampling distributions. For each ICD, data points come from 1000 independent simulations. [**B-D, F-H**] Quantifying the effect of equal [B-D] and adversarial [F-H] ICPs for different intensities (molecular count percentage with respect to baseline scenario: ICP = 0%), and varying number of perturbed cells for all different cell-fate categories (EPI, PRE, UND). For each fate category, color saturation indicates the fate error, i.e. the mean difference of cell-fate counts between the perturbed and unperturbed baseline systems. Each matrix entry (perturbation pair) is calculated from 1000 independent simulations.

We quantified the deviations from the typical ICD behavior using the relative Fate Error (FE), which measures the discrepancy in ICM specification accuracy for each cell fate (UND/EPI/PRE) between the perturbed scenarios and the reference (0% ICP) scenario at 48 hours; see also caption of Fig 9 for details about FE.

Our findings from both schemes indicate that when more than 25% of the cell population is affected by the ICD disturbances, the system experiences significant deviations in lineage distributions, regardless of perturbation strength. This suggests the existence of a critical threshold ratio of perturbed cells, beyond which the system's resilience is notably compromised. When this threshold is exceeded, profound gene expression imbalances emerge across the ICM population.

Both types of perturbation, whether implying scarcity or over-abundance of initial resources, result in similar FE values suggesting a potential correction mechanism that can overcome highly irregular cellular initial conditions. Notably, no significant deviation is observed when less than 25% of the cell population undergoes perturbation. Here the system demonstrates remarkable robustness, underscoring the tissue-level coordination inherent in the ICM specification process.

In perspective, testing the impact of restricting gene expression profiles for multiple cellular subpopulations of different sizes is also achievable via optogenetics [110, 111, 126]. Our approach predicts only minor deviations for final cell-lineage distributions, in terms of their accuracy and precision, when less than 25% of the initial cellular population is perturbed

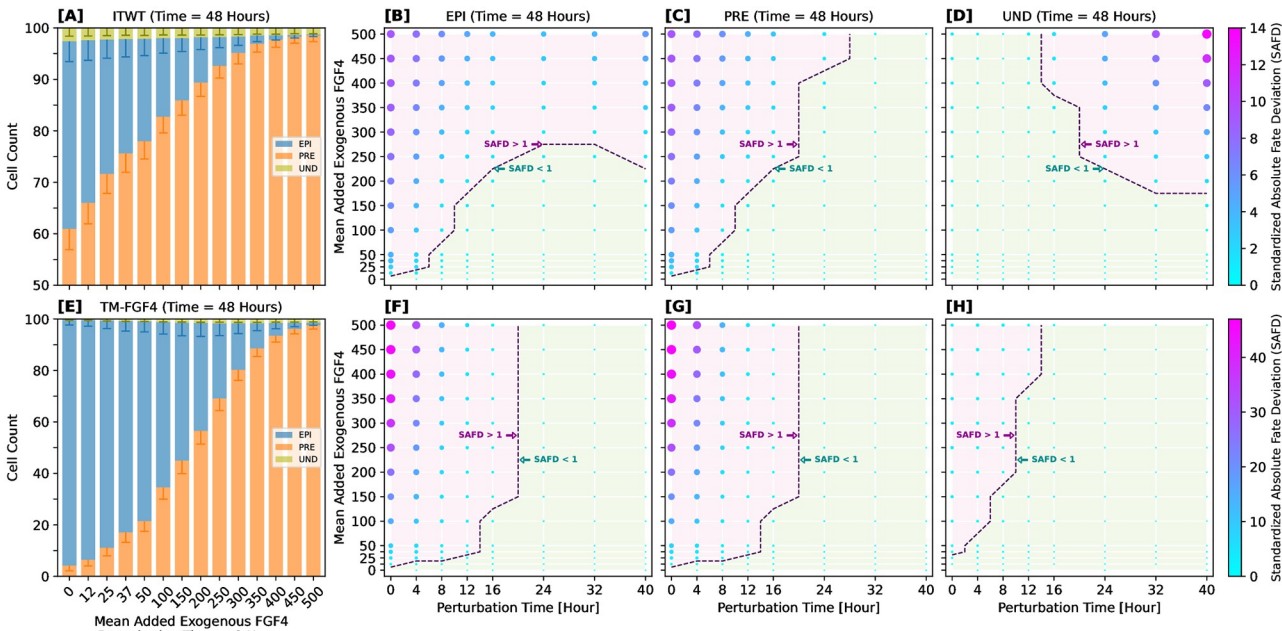

**Fig 10. Robustness to varying amounts of exogenous FGF4 added at distinct time points quantified by SAFD at 48 h: cell plasticity and FGF4 sensitivity are time-window-dependent.** [A] Final cell-fate composition (at 48 h) of the inferred-theoretical wild-type (ITWT) system, for all tested FGF4 protein perturbation magnitudes and FGF4 addition at simulation start (0 h). Bar height ranges from 50 to 100 cells. [B-D] Effect of different FGF4 perturbation magnitudes and FGF4 addition times on cell-fate composition at 48 h, quantified by Standardized Absolute Fate Deviation (SAFD) for all three fate categories (EPI, PRE, UND) in the ITWT system. SAFD (color represents amplitude) refers to the distance from the reference (or null) configuration (no exogenous FGF4 at 0 h) to every alternative perturbation magnitude and addition time. SAFD is defined as the absolute difference between the mean of reference and perturbed proportions, rescaling this absolute difference by the standard deviation of the reference proportion. The separating hyperplane indicates the boundary between the regions with distance $\leq 1$ and distance $> 1$. [E] Final cell-fate proportions (at 48 h) of the theoretical mutant system lacking FGF4 production (referred to as TM-FGF4), for all tested FGF4 protein perturbation magnitudes and FGF4 addition at simulation start (0 h). Bar height ranges from 0 to 100 cells. [F-H] Effect of different FGF4 perturbation magnitudes and FGF4 addition times on cell-fate composition at 48 h (quantified by SAFD) in the TM-FGF4 system. All bars and data points are calculated from separate batches of 1000 independent simulations for each perturbation. The following list represents a one-to-one correspondence between molecular concentrations and counts for mean added exogenous FGF4 protein, assuming a cellular culture medium with 100 compactly placed cells (concentrations are given in [ng/ml]): $\gamma$ = [0.0, 0.12, 0.25, 0.37, 0.5, 0.99, 1.49, 1.98, 2.48, 2.97, 3.47, 3.96, 4.46, 4.96]. Conversion formula: $\gamma = (x \cdot M)/(N_A \cdot V_{med})$. Where: $\gamma$ is the resulting mass concentration of FGF4 [ng/ml]; $x$ is the FGF4 copy number; $M$ is the molecular weight of FGF4 (25 kDa); $N_A$ is the Avogadro constant; $V_{med}$ is the medium's volume, $100 \cdot 4200$ µm³, as shown in Table 3.

accordingly; see Fig 9. Moreover, we predict that these deviations depend solely on the cumulative number of perturbed cells, rather than the specific locations of perturbation.

## Cell plasticity and FGF4-sensitivity are time-window dependent

Temporal modulation of FGF4 concentration and the corresponding shift in cell plasticity are key aspects of ICM cell differentiation, with numerous studies documenting their influence [7, 9, 21, 30, 128]. To examine whether our model can replicate the experimentally observed time-dependent responsiveness to FGF4 level changes, we introduced controlled perturbations of FGF4 in our simulations. This involved adding extra FGF4 molecules to each simulated cell at specific time points, mimicking the effect of exogenous FGF4.

We assessed the response to FGF4 perturbations in two models (ITWT and TM-FGF4) and at two distinct final simulation times, 48 and 96 hours. The second time point, while biologically irrelevant, was introduced to test whether manipulation of FGF4 levels can alter the typical time scale on which cell-fate commitment converges. Exogenous FGF4 addition was carried out at predetermined time points: simulation time = [0, 4, 8, 12, 16, 24, 32, 40] hours,

with the amount of added FGF4 molecules determined by a mix of Poissonian and binomial distributions (details in "Computational experiments"). Note that the TM-FGF4 system represents a theoretical mutant with blocked FGF4 production, mimicking a full loss-of-function phenotype for the *Fgf4* gene; this means that any FGF4 originates in these systems from the external addition.

We quantified the ICM patterning robustness to varying amounts of exogenous FGF4 using the Standardized Absolute Fate Deviation (SAFD), which measures the distance, in terms of mean cell-fate proportions, from the null configuration (no exogenous FGF4 at 0 hours) to every alternative perturbation-magnitude-and-addition-time configuration; see also caption of Fig 10 for details about SAFD.

Our analysis at the 48-hour mark shows that the TM-FGF4 model is highly responsive to exogenous FGF4 added at the start of the simulation (addition time = 0 hours). Depending on the FGF4 count, it is possible to manipulate the lineage ratios and rescue the PRE fate (Fig 10E), which is in line with recent in vitro experiments [9]. However, we identify an end-of-plasticity time point around 24 hours, after which the system becomes insensitive to additional FGF4 and locks into its pre-existing cell lineage proportions (Fig 10F–10H). In the ITWT system, cell fate ratios can also be manipulated by varying the FGF4 amount, although no distinct end-of-plasticity time point is observed in this model (Fig 10A–10D).

The quantitative predictions in Fig 10A–10D and 10E–10H motivate potential in-vitro experiments. These predictions not only formulate *a priori* expectations for the robustness of the real system, but also for its response to varying amounts of exogenously applied FGF4. To our knowledge, while similar experiments with limited scope have been performed for an in-vitro mutant system lacking FGF4 production [9], there is currently no comparable experiment akin to Fig 10A–10D for a wild-type-like ICM system.

To ease the translational value of our computational FGF4-based perturbations, we supply a set of quantities that directly translate the numbers in Fig 10 to wet-lab experimental settings; see also caption of Fig 10. The following list represents a one-to-one correspondence between molecular concentrations and counts (as seen in Fig 10) for mean added exogenous FGF4 protein. For simplicity, we assume a cellular culture medium with 100 compactly placed cells, and the concentrations are given in [ng/ml]:

$\gamma$ = [0.0, 0.12, 0.25, 0.37, 0.5, 0.99, 1.49, 1.98, 2.48, 2.97, 3.47, 3.96, 4.46, 4.96]

The employed conversion formula for these predictions is as follows: $\gamma = (x \cdot M)/(N_A \cdot V_{med})$. Where $\gamma$ is the resulting mass concentration of FGF4 [ng/ml]; $x$ is the FGF4 copy number, as shown in Fig 10; $M$ is the molecular weight of FGF4 (25 kDa); $N_A$ is the Avogadro constant; $V_{med}$ is the medium's volume, $100 \cdot 4200 \ \mu m^3$, as shown in Table 3.

We remark that Fig 10E recapitulates the experimental observation shown in Fig 2F of [9]. Notably, our simulation data match the order of magnitude for their experimental measurements. While our values are slightly smaller than the ones reported in [9], this can be explained by the fact that we consider a closed system from which FGF4 cannot leak out by diffusion.

Extending the simulation to 96 hours, the TM-FGF4 exhibits similar behavior as in the 48-hour case (Fig 11E–11H). However, the ITWT system now displays a gradual loss of cell plasticity; beyond the 32-hour mark, additional FGF4 does not significantly alter final fate ratios, provided the average amount of added FGF4 remains below 250 molecules per cell (Fig 11A–11D).

In summary, our simulations with exogenously administered FGF4 underscore that ICM cell plasticity is confined to a specific time window. The ICM population's transient sensitivity to external FGF4 allows for the maintenance of EPI and PRE lineage proportions under normal conditions. Nevertheless, the balance between these two fates can be influenced by the

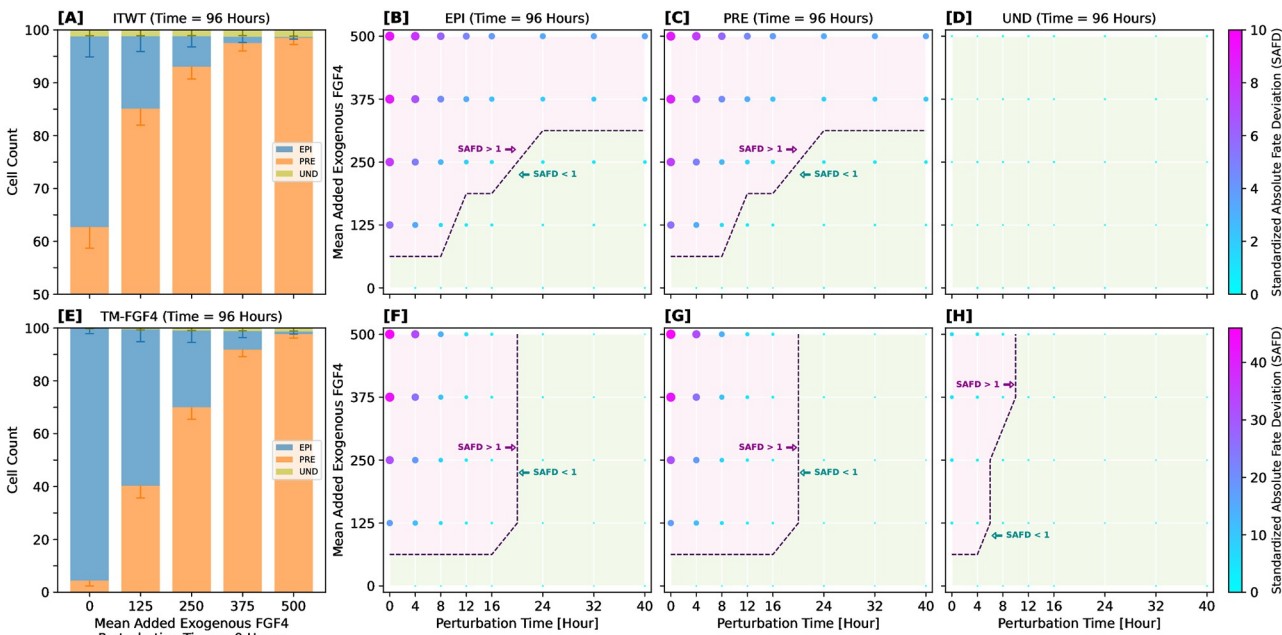

**Fig 11. Robustness to varying amounts of exogenous FGF4 added at distinct time points quantified by SAFD at 96 h: Cell-fate composition perturbed by non-endogenous FGF4 addition remains largely unchanged beyond 48 h.** [**A**] Final cell-fate composition (at 96 h) of the inferred-theoretical wild-type (ITWT) system, for all tested FGF4 protein perturbation magnitudes and FGF4 addition at simulation start (0 h). Bar height ranges from 50 to 100 cells. [**B-D**] Effect of different FGF4 perturbation magnitudes and FGF4 addition times on cell-fate composition at 96 h (quantified by SAFD) in the ITWT system. [**E**] Final cell-fate proportions (at 96 h) of the theoretical mutant system lacking FGF4 production (referred to as TM-FGF4), for all tested FGF4 protein perturbation magnitudes and FGF4 addition at simulation start (0 h). Bar height ranges from 0 to 100 cells. [**F-H**] Effect of different FGF4 perturbation magnitudes and FGF4 addition times on cell-fate composition at 96 h (quantified by SAFD) in the TM-FGF4 system. See caption of Fig 10 for definition of SAFD.

timing and concentration of exogenous FGF4, showcasing the nuanced interplay between external factors and intrinsic developmental processes.

## Discussion

The specification from the inner cell mass (ICM) to epiblast (EPI) and primitive endoderm (PRE) lineages is a pivotal process in preimplantation blastocyst formation, representing a key paradigm in mammalian tissue development. This process exemplifies self-organization, balancing high plasticity with strong robustness of fate proportioning [2]. Its reproducibility and adaptability to experimental perturbations are conserved features across mammalian species [5], highlighting its importance for studying the emergence of pluripotency and homeostasis, especially within the context of embryonic stem cell (ESC) cultures [7].

ICM-derived ESCs possess the remarkable ability to form any germ layer. But, despite their self-renewal, regular developmental timing, and seemingly deterministic tissue patterning capabilities, ESC populations display significant plastic heterogeneity [23, 29, 30, 72]. This property reflects the intrinsically stochastic expression of key gene regulatory factors, the relatively small number of blastocyst cells, as well as the spatial variability of intercellular signaling sources [5, 7, 129]. With the advent of sophisticated genome manipulation tools and high-throughput screening techniques such as scRNA-seq [11, 130], mouse experimental models have advanced our mechanistic understanding of diseases such as cancer, guiding both drug

design and disease etiology, thereby substantiating their high relevance to human medicine [6, 7]. Although extrapolating findings from animal studies to human biology remains challenging, theoretical models facilitate the investigation of patterning principles through unifying mathematical themes, elucidating universal properties, and predicting novel experiments across paradigmatic developmental biology systems [88, 131–134].

However, existing theoretical models of mouse ICM fate specification exhibit a fundamental conceptual gap due to the lack of detailed mechanistic understanding of the dynamical landscape of cellular potency and plasticity, as they are largely phenomenological and primarily deterministic in nature [8, 15, 22, 38]. As such, these approaches fail to rigorously quantify the implications of noise emerging from the basic biochemical processes driving this developmental process.

To correctly capture the inherent randomness in ICM differentiation, we constructed a biophysics-rooted spatial-stochastic gene-regulation model. This model is simulated via the Reaction-Diffusion Master Equation (RDME) formalism, and we embedded it into a Simulation-Based Inference (SBI) framework, capitalizing on recent advancements in Machine Learning (ML). This combined workflow enables a mechanistic description of the ICM patterning dynamics in multi-cellular settings, using realistic lifetimes for the involved molecular species. It also offers a biophysics-grounded implementation of the mesoscopic processes generating biochemical noise at both cell and tissue scales.

Leveraging this combined workflow, we developed the Inferred-Theoretical Wild-Type (ITWT) model, which collectively recapitulates key experimental findings: (1) indispensability of FGF4-mediated signaling for proper ICM patterning [30, 99], evidenced through the FGF4-coordinated stimulation of the PRE fate that requires prior emergence of the FGF4-producing EPI lineage; (2) high reproducibility and robustness of EPI-PRE lineage proportions, irrespective of the cell number (system size) and despite the narrow timescale of blastocyst formation (1.5–2 days of embryonic development) [38, 99]; (3) significant ICM sensitivity to exogenously applied FGF4 in mutant-like conditions [9], with ICM plasticity being adjustable based on application time and dosage strength. Importantly, our computational approach produces quantitative predictions that could inform future experiments: (4) intercellular communication via FGF4 can functionally improve ICM differentiation robustness, reducing fate proportioning error by 10–20% compared to a purely cell-autonomous system; (5) increased variability in the initial conditions of key cellular resources can enhance the accuracy while sustaining precision for EPI-PRE cell-type proportioning, with fate distributions remaining robust when less than 25% of the cell population starts with perturbed initial conditions.

In the absence of FGF4 coupling, the simulated blastocyst fails to establish correct cell fate proportions, displaying a strong bias towards the EPI fate. We thus argue that, given a default naive pluripotent state in ICM-like systems, successful cell-fate specification necessitates a tissue-level mechanism that orchestrates the emergence of distinct cellular fates by providing coordinating feedback between cells with distinct plasticity potentials.

Previous experimental and simulation results also underscore the importance of cell-cell signaling in maintaining reproducible lineage proportions globally while facilitating correct pattern formation locally [8, 9, 35]. Interestingly, our analysis of cell neighborhood composition reveals that the communication range of FGF4, though essentially limited to nearest neighbors, suffices for ensuring effective signaling. This observation aligns with recent findings from both in-vitro and in-silico studies [9, 85], highlighting the nuanced roles of local signaling dynamics in complex tissue patterning.

We successfully recover the temporal sensitivity of the ICM to exogenous FGF4 for a mutant system lacking FGF4 production. We observe a specific time window during which

the ICM can respond to external FGF4, with the ability to adjust lineage proportions depending on the timing and dosage of the addition. This finding aligns with recent experimental observations [9] and highlights the importance of timing in developmental processes. In addition, we go beyond the current experimental data, providing quantitative predictions for multiple application times and dosage strengths of exogenous FGF4 signaling molecules, not only for mutant-like conditions but also for our wild-type-like system (ITWT).

We find that system size (number of cells) does not significantly influence the accuracy of attaining the correct lineage proportions. This is in line with previous studies reporting that the mouse blastocyst exhibits resilience to ICM size variations, maintaining consistent patterning irrespective of cell number [1, 8]. However, we demonstrate that the precision of cell specification is system-size dependent. Our results predict that cell-fate misspecification can be reduced down to $\sim 10\%$ when the system size surpasses $\sim 50$ cells, which is comparable to the typical number of cells in the ICM around E3.5.

Notably, our simulations show that increased variability in initial conditions at the cellular level does not necessarily constitute a detriment for the tissue-level dynamics. Instead, increased variability in key initial molecular resources, if not excessive, can enhance the accuracy while sustaining the precision of cell-fate specification. Moreover, our simulations predict no significant deviations for final cell-lineage distributions, with respect to accuracy and precision, when less than 25% of the cellular population experiences initial condition perturbations, regardless of perturbation strength and biases.

A paramount challenge in biophysical mechanistic modeling is the estimation of parameter values allowing the constructed model to faithfully recapitulate the characteristics of the considered biological system. This task becomes particularly complex for spatial-stochastic and mechanistic models due to the need for analyzing behavior across numerous independent simulation samples, significantly increasing the computational demands for navigating their vast parameter spaces [92, 135, 136]. In response to this challenge, our approach integrates an AI-powered Simulation-Based Inference (SBI) method with traditional ML techniques, specifically employing the Sequential Neural Posterior Estimation (SNPE) algorithm [86, 87, 89]. This strategy leverages simulation data to efficiently traverse parameter space, incorporating both direct analysis of simulation outcomes and qualitative observations to identify parameter distributions that align with the expected behaviors of the system, encoded in high-level, low-dimensional utility functions ("target scores").

We utilized a state-of-the-art SBI toolbox [137], which facilitated the integration of the SNPE algorithm into our workflow. This allowed us to train artificial neural networks (ANNs) for predicting model parameter sets capable of reproducing the targeted ICM patterning behavior of several model variants, corresponding to both wild-type and mutant systems. These predictions are point and interval estimates of biophysical model parameters, which could be experimentally tested or used as guidance in future approaches. For example, Table 4 and S2 Fig shows full predictive ranges for the effective lifetimes of both cytoplasmic FGF4 protein and the membrane-bound FGFR1-FGF4 (receptor-ligand) complex.

We find that the lifetime of intracellular FGF4 does not need to be tightly regulated, as its functional role is part of the extracellular feedback mechanism. To test the implications of manipulating FGF4 lifetime for the regulation of cell-fate decisions, it is conceivable to employ pharmacological tools such as protein therapeutics or targeted enzymatic actions [138, 139]. In contrast, the lifetime of the FGFR1-FGF4 complex needs to be tightly regulated to achieve ideal cell-type proportioning behavior. This prediction could potentially be tested using FRET (Förster Resonance Energy Transfer) experiments [140], to measure or estimate the required fine-tuning of the stability of the complex.

Our methodology imposes minimal constraints on the inference problem by leveraging only essential experimental observations. This strategy prevents model overfitting, allowing for the extrapolation of system behaviours spanning multiple spatial and temporal scales not directly observed in experimental data. With this approach, we underscore the distinctions between ML models, which prioritize universal prediction at the expense of modeling interpretability, and mechanistic models, which focus on exploratory hypotheses to uncover causal relationships at the expense of modeling fidelity [88, 141]. By merging these paradigms, we demonstrate that despite the scarcity of detailed quantitative experimental measurements, the flexibility and predictive capabilities of ANNs can aid generating full-featured quantitative predictions by imposing key empirical qualitative observations to mechanistic biophysical models. To our knowledge, our work constitutes the first application of an AI-powered SBI framework to spatial-stochastic predictive modeling in developmental biology.

While here we focused on a minimal spatial geometry for targeted assessment of the inter-play between biochemical stochasticity and spatial coupling, ICM development is influenced by important additional factors, such as cell divisions and force interaction among cells. Future elaborations of our framework will incorporate suitable tissue-scale dynamics, which will integrate the stochastic dynamics of single-cell gene expression and inter-cellular signaling with the constant remodeling of the tissue geometry. Several approaches addressing tissue remodeling problems have already been proposed; thus, we will focus on exploiting several of these computational platforms, such as the ones described in [142] and [25], suitable for exploring shape homeostasis emergence in simple 3D systems. This will enable the study of how cell neighborhoods varying both in time and space influence ICM lineage differentiation, while leveraging recently recorded tissue structural data [10, 35, 37, 119].

The extended framework will also feature the other two important constitutive elements of the developing mouse blastocyst, namely the blastocoel (blastocyst cavity) and the trophectoderm (TE), exposing an interesting research direction, as recent experimental evidence suggests that the expansion of the mouse blastocyst lumen could play a role in stimulating ICM fate differentiation. This is thought to occur through an interplay of mechanical clues and position-specific induction of gene-expression, possibly mediated by FGF4 molecules deposited in the blastocoel [28, 143].

Perspectively, our AI-powered approach provides a promising basis for establishing virtual replicas, or *digital twins*, of early embryonic development. The generative model capability of our approach facilitates the discovery of complex parameter space structures and compensation mechanisms, akin to recent advancements in fields such as structural biology [144, 145] and neuroscience [146, 147]. This has potential benefits for the study of human in-vitro fertilization and treatment of prevalent gestational complications [5], as it could circumvent the need to establish associated experimental systems, which may come with ethical restrictions. The highly detailed synthetic trajectories produced by our simulation framework could be coupled with future experiments combining, for example, both scRNA-seq and smFISH (single-molecule fluorescence in situ hybridization) technologies [76], in order to disentangle the noisy dynamics at transcription and translation scales for individual genes by exploiting computer-assisted insights. This outlook also underlines the importance of identifying the essential noise-control mechanisms that maintain a well-balanced ratio between EPI and PRE populations, given that breaking this balance can have significant physiological ramifications for the postimplantation embryo [3, 99].

In conclusion, our AI-parameterized model underscores the complexity and robustness of the EPI-PRE lineage specification, generating unique insights into the interplay of stochasticity, tissue-level signaling, feedback mechanisms, and system size (number of cells) in ICM development. These findings not only deepen our understanding of the developing early

mouse embryo under genuine biochemical noise conditions but also provide a comprehensive framework for exploring similar controlled stochastic processes in related biological systems. Potential examples include, but are not limited to, the human blastocyst formation [2, 6, 7, 148], the *Bacillus subtilis* competence circuit [149–151], and the *Dictyostelium discoideum* cell-type proportioning [152].

## Materials and methods

### Computational model of mouse blastocyst (ICM cell differentiation)

The model comprises two fundamental building blocks. The first submodel (cell level) consists of the GRN (NANOG-GATA6-FGF4) coordinating the ICM cell specification process. The second submodel (tissue level) describes the cell-cell signaling dynamics. Unlike other existing models [11, 15, 22, 23], our modeling approach does not integrate the notion of noise as a purely extrinsic component. Instead of an arbitrary noise source, we employ a mesoscale description which incorporates noise as an intrinsic component. Thus, noise plays an essential role for faithfully simulating the temporal evolution of our biological system model.

Indeed, the presence of noise in biophysical models is deemed central for discerning the main features of gene regulatory processes [82, 153, 154]. Conventionally, noise is separated into intrinsic and extrinsic categories [121, 122]. While it is problematic to give a clear delimitation of these two categories, here we provide a general interpretation of their scope within the context of our study.

Intrinsic noise arises from the nature of biochemical reaction and diffusion events; i.e., discrete molecules randomly diffuse and randomly react when a collision occurs between each other. As such, intrinsic noise commonly refers to local fluctuations within basic gene regulation mechanisms; e.g., transcription and translation. Extrinsic noise originates from cellular environment variations or changes. Hence, extrinsic noise typically alludes to global factors systematically affecting all cells but irregularly propagating across cellular mechanisms; for example, cell cycle timing and cellular resource partitioning. Nevertheless, recent experimental and theoretical works argue for treating both noise categories as inseparable entities [155–158].

When there is a large number of molecules at play, a biochemical dynamics model typically follows a deterministic formulation: reaction rates are represented by constant functions, species amounts are represented by concentrations (continuous-time functions), and it primarily follows an ordinary differential equation (ODE) scheme. By contrast, when there is a small number of molecules at play, a stochastic formalism takes precedence.

Generally, stochastic biochemical dynamics models are formulated as continuous-time Markov chains (CTMCs); i.e., continuous-time discrete state-space Markov processes. Numerous mathematical and computational methods have been developed for analysis and simulation of such stochastic formulations [159–166]. These techniques methodically incorporate stochasticity, which is relevant for understanding the effects of noise on cell-cell variability.

A biochemical reaction network involves multiple reactions (edges) and species (vertices or nodes); a CTMC is the most common model of such a network. Particularly, biophysical systems can be abstracted using the Chemical Master Equation (CME) formalism; Eq (1) [166, 167].

$$\frac{\partial P}{\partial t}(\boldsymbol{x}, t \,|\, \boldsymbol{x}_0, t_0) = \sum_{j=1}^{M} a_j(\boldsymbol{x} - \boldsymbol{v}_j, t) P(\boldsymbol{x} - \boldsymbol{v}_j, t \,|\, \boldsymbol{x}_0, t_0) - a_j(\boldsymbol{x}, t) P(\boldsymbol{x}, t \,|\, \boldsymbol{x}_0, t_0) \tag{1}$$

Where: $\boldsymbol{x}$ is the state vector of the system $\boldsymbol{X}$ (CTMC); $\boldsymbol{x} = \boldsymbol{X}(t) = [X_1(t), \ldots, X_N(t)]$; there are $N$

biochemical species ($i \in \{1, \ldots, N\}$); each entry of $\boldsymbol{x}$ represents the copy number of a given biochemical species $S_i$; $P(\boldsymbol{x}, t | \boldsymbol{x}_0, t_0)$ is the time-dependent probability density function of $\boldsymbol{x}$; $\boldsymbol{x}_0$ is the initial state vector; $t_0$ is the initial time; there are $M$ reactions ($j \in \{1, \ldots, M\}$); $a_j$ is the nominal rate of reaction $R_j$; $v_j$ is the state-change vector (set of stoichiometric coefficients) of reaction $R_j$; $a_j(\boldsymbol{x}, t)$ is the propensity function (effective rate) of reaction $R_j$ when the system $\boldsymbol{X}$ is in state $\boldsymbol{x}$ at time $t$.

The full GRN implemented by our simulator includes all the molecular species relevant for the developmental system dynamics, together with several auxiliary (computational) species; this procedure facilitates the inclusion of all important molecular relationships and the tracking of crucial model variables. An extensive list of species, relations, and nomenclature guidelines is available from the corresponding simulator scripts. For ease of exposition, Table 1 presents only the actual biochemical species considered for our model.

Within the CME framework, a well-mixed or reaction-limited system is the main assumption; i.e., molecular diffusion is relatively fast compared to the speed of any biochemical reaction. The most popular method to simulate models following the CME formulation is the Stochastic Simulation Algorithm (SSA), a scheme introduced and rigorously proven to be physically relevant by the late Daniel T. Gillespie [159, 168].

Correspondingly, molecular diffusion speed can guide the choice of a biochemical dynamics representation. Fast diffusion is synonym with spatially-uniform distribution of resources; a well-mixed or homogeneous environment. Slow diffusion is synonym with spatial correlation and other spatial factors, which creates a heterogeneous environment.

While there exist multiple techniques tackling different spatial and temporal scales [163, 169–172], we aimed for balance between computational efficiency and biophysical realism. Consequently, we have followed the formalism of the Reaction-Diffusion Master Equation

**Table 1. Summary of biochemical species.**

| Name | Description |
|---|---|
| *Nanog* | {Gene, Promoter} |
| *Gata6* | {Gene, Promoter} |
| *Fgf4* | {Gene, Promoter} |
| *Nanog* mRNA | Messenger RNA |
| *Gata6* mRNA | Messenger RNA |
| *Fgf4* mRNA | Messenger RNA |
| NANOG | Protein |
| P-NANOG | {Protein, Phosphorylated NANOG} |
| GATA6 | Protein |
| FGF4 | Protein |
| FGFR | {Protein, Fibroblast Growth Factor Receptor (1/2)} |
| M-FGFR-FGF4 | {Protein, FGFR-FGF4 Monomer Complex} |
| D-FGFR-FGF4 | {Protein, FGFR-FGF4 Dimer Complex} |
| I-ERK | {Protein, Inactive ERK (1/2)} |
| A-ERK | {Protein, Active ERK (1/2)} |

Note. Our computational model has more than 50 species, but we only introduce in this table the species directly related to the biological problem. The other purely-computational species are necessary for correctly analyzing and tracing the complex GRN simulated dynamics.

(RDME); Eq (2) [173, 174].

$$\frac{\partial P}{\partial t}(\boldsymbol{x}, t \mid \boldsymbol{x}_0, t_0) = \mathcal{R}P(\boldsymbol{x}, t \mid \boldsymbol{x}_0, t_0) + \mathcal{D}P(\boldsymbol{x}, t \mid \boldsymbol{x}_0, t_0)$$

$$\mathcal{R}P(\boldsymbol{x}, t \mid \boldsymbol{x}_0, t_0) = \sum_{j=1}^{M}\sum_{k=1}^{L} a_j(\boldsymbol{x}_k - \boldsymbol{v}_j, t)P(\boldsymbol{x}_1, \ldots, \boldsymbol{x}_k - \boldsymbol{v}_j, \ldots, \boldsymbol{x}_N, t \mid \boldsymbol{x}_0, t_0)$$

$$- a_j(\boldsymbol{x}_k, t)P(\boldsymbol{x}, t \mid \boldsymbol{x}_0, t_0) \tag{2}$$

$$\mathcal{D}P(\boldsymbol{x}, t \mid \boldsymbol{x}_0, t_0) = \sum_{i=1}^{N}\sum_{k=1}^{L}\sum_{h=1}^{L} P(\boldsymbol{x}_1, \ldots, \boldsymbol{x}_{ik} - \boldsymbol{w}_{ikh}, \ldots, \boldsymbol{x}_N, t \mid \boldsymbol{x}_0, t_0)$$

$$\times b_{ikh}(\boldsymbol{x} - \boldsymbol{w}_{ikh}, t) - P(\boldsymbol{x}, t \mid \boldsymbol{x}_0, t_0) \times b_{ikh}(\boldsymbol{x}_{ik}, t)$$

Where: $\mathcal{R}P(\boldsymbol{x}, t \mid \boldsymbol{x}_0, t_0)$ and $\mathcal{D}P(\boldsymbol{x}, t \mid \boldsymbol{x}_0, t_0)$ are the reaction and diffusion components of the equation, respectively; $\boldsymbol{x}_k$ (or $\boldsymbol{x}_h$) is the state vector of the voxel $\Omega_k$ (or $\Omega_h$); $\boldsymbol{x}_{ik}$ (or $\boldsymbol{x}_{ih}$) is the copy number of $S_i$ for $\Omega_k$ (or $\Omega_h$); there are $L$ voxels ($k, h \in \{1, \ldots, L\}$); $b_{ikh}$ is the nominal diffusion rate of $S_i$ from $\Omega_k$ to $\Omega_h$; $w_{ikh}$ is the state-change vector (set of stoichiometric coefficients) for the diffusion of $S_i$ from $\Omega_k$ to $\Omega_h$.

The RDME framework works at the mesoscopic level, and its simulation schemes are based on custom versions of the SSA tailored to incorporate reaction-diffusion processes. Here, we depart from one of such schemes, the Next Subvolume Method (NSM) [165, 175], which separates events into two distinct kinds: reaction firing inside every cell, and diffusive jumps between cells. For our system model, each cell is treated as a well-mixed voxel/environment, and tissue communication materializes by representing signaling-molecule diffusion as a morphogen-exchange process between neighboring cells; as such, we will commonly refer to this process simply as "diffusive jump", "jump diffuse", or "jump diffusion".

To put it briefly, our event-driven simulator is congruent with the NSM because it involves the SSA, the computational spatial domain is partitioned into artificially well-mixed compartments where only molecules belonging to the same compartment can react, diffusive jumps transport molecules between neighboring voxels, and there are well-defined event queues. Outside these shared features, our simulator allows for complex interactions among voxels or cells, which facilitates the presence of multiple tissue types and the corresponding relabeling of molecules once they undergo jump-diffuse steps. Likewise, the nominal diffusive-jump rates are calculated based on an arbitrary system model geometry and the principle of conservation of (molecular) flow; unlike the NSM, which calculates the nominal jump-diffuse rates based on a regular cubic geometry and the voxel size.

**Key stages of mouse embryo preimplantation development.** To guide our model construction and in silico analysis, we relied on wet-lab experimental descriptions of core phases in early mouse development. The mouse preimplantation period encompasses a series of morphological and molecular changes which transform the zygote (one totipotent cell) into an approximately 256-cell (7–8 cleavages) embryo at around E4.5; at this point, the embryo comprises three spatially segregated cell types: TE, EPI, and PRE. For a complete recap of the mouse embryo preimplantation development, please see [2, 3, 29].

The first cell-fate decision happens between E2.5 and E3.0 (from 8- to 32-cell stage): cells acquire TE or ICM identities. The second cell-fate decision happens at the ICM between E3.0 and E4.0 (from 32- to 128-cell stage). From E4.0 to 4.5, EPI and PRE populations spatially separate. While it is customary to define the blastocyst-formation period between E3.0 and E4.5 [1, 30], these boundaries are ultimately arbitrary as development occurs in a continuum and

diverse experimental arrangements/conditions are in use between distinct labs. Moreover, ICM cells adopt their next identities asynchronously as the blastocyst forms [21, 29, 99]. Together with these aspects, it is also commonly accepted that cells are already coexpressing *Nanog*- and *Gata6*-related factors at around E2.75 [2, 176], plus *Fgf4* expression is already perceptible at around E3.25 [1, 30]. For these reasons, our standard model simulations target a time-window of 48 hours (E2.75-E4.75); this range allows us to circumvent potential discrepancies among timing annotations and keep a temporally faithful description of the biological system under study.

**Fundamental interactions among central GRN components.**   Many processes coexist during blastocyst development. These processes materialize at multiple temporal/spatial scales and embody the relationships of numerous components operating simultaneously. A vast number of elements conjointly orchestrate developmental progress scaling from cell-level adaptable gene expression mechanisms to tissue-level mechanical/signaling coordination structures. Particularly, the GRN controlling the ICM specification process has a rich collection of components and interactions. Here, we model this GRN by accounting for the key interactions among its main components, as reported by recent experimental studies.

To start, we suppose that our core GRN motif consists of the species and interactions primarily governing the *Nanog*- and *Gata6*-gene expression dynamics. This collection of ingredients only includes *Nanog* mRNA, *Gata6* mRNA, NANOG protein, and GATA6 protein, naturally. As transcription factors (TFs), both NANOG and GATA6 proteins exhibit self-activation and mutual repression [3, 73].

The remainder of the complete GRN encompasses all the species and interactions secondarily governing the *Nanog*- and *Gata6*-gene expression dynamics. This group includes *Fgf4* mRNA, FGF4 protein, and ERK protein. Among these explicit elements, we also implicitly include two FGF receptor (FGFR) complexes, which concertedly facilitate biochemical signal transduction during blastocyst formation [67, 177]. Recently, a comprehensive experimental study demonstrated that NANOG and GATA6 proteins are capable of jointly binding to both EPI and PRE *cis*-regulatory modules [73]. This concrete evidence supports the previously proposed direct NANOG activation plus GATA6 repression of the *Fgf4* gene, both in vitro and in vivo [15, 117]. Likewise, ERK has been indicated to play a crucial role for this GRN [7, 178]. At transcriptional level, ERK is capable of recruiting diverse repressor TFs to *Nanog*-gene loci [105]. For antisymmetry and simplicity, we assumed that ERK is capable of recruiting diverse activator TFs to *Gata6*-gene loci; however, there is indeed some experimental evidence indicating such a motif [102]. At post-translational level, NANOG phosphorylation by ERK promotes its instability, which consequently reduces its lifetime [105, 113, 179]. Contrastively, it has been reported that GATA6 phosphorylation by ERK enhances its stability; nevertheless, the implications of this motif are not completely clear and we exclude it [102].

Finally, the upstream release of FGF4 induces the FGFR-FGF4 monomer complex formation, which successively induces the FGFR-FGF4 dimer complex formation. This FGFR-FGF4 dimer complex ultimately triggers the pathways downstream of ERK [7, 114, 128, 178].

**Model at cell scale.**   The core GRN motif is exclusively comprised by *Nanog*- and *Gata6*-related elements. To be more precise, all their directly related species and interactions. The rest of the full GRN is built around the core motif, thus consolidating the remaining elements and their collective effects on the dynamics of the two main players. Importantly, we have arranged all cell-scale reaction events into several groups as follows: summary of gene expression dynamics; promoter binding and unbinding; mRNA synthesis and degradation; protein synthesis and degradation; FGFR activation and inactivation; ERK activation and inactivation; NANOG phosphorylation and dephosphorylation. In that regard, we report all the particular interactions implemented by our simulator and their respective literature sources.

**Table 2. Regulation of gene transcription.**

| Gene (Promoter) | TF | TFBSs | TF Role | PVSs |
|---|---|---|---|---|
| *Nanog* | {NANOG, P-NANOG} | 4 | Activator | [15, 38] |
| *Nanog* | GATA6 | 4 | Repressor | [15, 38] |
| *Gata6* | {NANOG, P-NANOG} | 4 | Repressor | [15, 38] |
| *Gata6* | GATA6 | 4 | Activator | [15, 38] |
| *Fgf4* | {NANOG, P-NANOG} | 2 | Activator | Self |
| *Fgf4* | GATA6 | 2 | Repressor | Self |
| *Nanog* | A-ERK | 3 | Repressor | [15, 38] |
| *Gata6* | A-ERK | 3 | Activator | [15, 38] |

Notation: TF = Transcription Factor; TFBSs = TF Binding Sites; P-NANOG = Phosphorylated NANOG; A-ERK = Active ERK; PVSs = Parameter Value Sources.

**Summary of gene expression dynamics.** The only three genes with an explicit mRNA step are *Nanog*, *Gata6*, and *Fgf4*; Table 2 summarizes their relationships. The expression of the other two genes (*Fgfr* and *Erk*) is only visible either at an implicit form or at the protein level. FGFR also does not have an explicit protein count as it is available rather uniformly on the cell membrane [67, 140, 177, 180, 181]; instead, FGFR appears as an implicit component of the auxiliary protein variables/species M-FGFR-FGF4 (FGFR-FGF4 monomer complex) and D-FGFR-FGF4 (FGFR-FGF4 dimer complex), which helps reducing the number of reactions as well as alleviating the computational resources. ERK has itself two different protein forms: I-ERK (inactive ERK) is abundant in the cell cytoplasm [128, 182], and it is already present at the start of all the simulations; A-ERK (active ERK) is always inversely proportional to I-ERK, thus it is a product of the action of D-FGFR-FGF4 on I-ERK.

**Promoter binding and unbinding.** Each of the three genes *Nanog*, *Gata6*, and *Fgf4* has a respective promoter with multiple independent binding sites for each of its TFs; check Table 3. Both gene activation and repression are cooperative: exactly $q$ TF copies must be simultaneously bound to their particular promoter sites for activation or repression of expression; by default, repression takes precedence over activation. For a given TF $A$, the (time-dependent) effective promoter binding rate is calculated via the formula $\tilde{k}_b = 4\pi dDA_t/V$ (diffusion-limited regime). Here, $d = 10$ nm is a typical binding-site diameter [46], $D = 10$ μm$^2$s$^{-1}$ is the cytoplasmic/nuclear TF diffusion coefficient, $A_t$ is the TF copy number at time $t$, and $V = 4200$ μm$^3$ is a typical mouse blastocyst cell volume [183–185]. For our system model, we do not know the diffusion coefficients of all the biochemical species, thereby we simply made an educated guess and assumed the same value for all the TFs based on other representative biological systems [46, 167, 186, 187]. Concisely, the nominal promoter binding rate is determined by the equation $k_b = \tilde{k}_b/A_t = 4\pi dD/V$.

We model TF cooperativity by expressly tuning the promoter unbinding rates. This rate tuning influences the promoter regulation model to mimic a Hill-function-like (nonlinear) transcriptional response. The usage of the Hill function is a staple of phenomenological modelling, however it is incompatible with mechanistic modelling; directly using a Hill equation as a reaction propensity function ignores the non-instantaneous (stochastic) nature of delays between biochemical events and introduces several other simulation artifacts [188–190]. We use the concepts of a half-saturation constant and a cooperativity coefficient to perform promoter unbinding rate tuning; accordingly, both quantities are incorporated into elementary reactions to describe promoter unbinding dynamics. This half-saturation constant ($h_{act}$ or $h_{rep}$ depending on TF role) is a free model parameter which dictates the threshold of TF copies

**Table 3. Summary of fixed model parameters.**

| Name | Alias | Description | Value |
|---|---|---|---|
| $V$ | | Cell volume | $4200\ \mu m^3$ |
| $D$ | | Protein diffusion coefficient (cytoplasm or nucleus) | $10\ \mu m^2 s^{-1}$ |
| $d$ | | Promoter binding-site diameter | 10 nm |
| $k_b$ | | Transcription factor binding rate | $0.3 \cdot 10^{-3}\ s^{-1}$ |
| $k_{coop}$ | | Cooperativity coefficient | 5 |
| $c_{basal}$ | | Relative contribution of basal mRNA production | 0.2 |
| $c_{find}$ | | Relative contribution of full-induction mRNA production | 0.8 |
| $\overline{M}_{Nanog}$ | | Mean steady-state mRNA copy number at full induction | 250 copies |
| $\overline{M}_{Gata6}$ | | Mean steady-state mRNA copy number at full induction | 250 copies |
| $\overline{M}_{Fgf4}$ | | Mean steady-state mRNA copy number at full induction | 200 copies |
| $\tau_{m,Nanog}$ | $t_{\frac{1}{2},m,Nanog}$ | Lifetime (or half-life) for a molecule of mRNA | 4 hours |
| $\tau_{m,Gata6}$ | $t_{\frac{1}{2},m,Gata6}$ | Lifetime (or half-life) for a molecule of mRNA | 4 hours |
| $\tau_{m,Fgf4}$ | $t_{\frac{1}{2},m,Fgf4}$ | Lifetime (or half-life) for a molecule of mRNA | 4 hours |
| $\overline{P}_{NANOG}$ | | Mean steady-state protein copy number at full induction | 1000 copies |
| $\overline{P}_{GATA6}$ | | Mean steady-state protein copy number at full induction | 1000 copies |
| $\overline{P}_{FGF4}$ | | Mean steady-state protein copy number at full induction | 800 copies |
| $\overline{P}_{ERK}$ | | Mean steady-state protein copy number | 1000 copies |
| $\tau_{p,NANOG}$ | $t_{\frac{1}{2},p,NANOG}$ | Lifetime (or half-life) for a molecule of protein | 2 hours |
| $\tau_{p,GATA6}$ | $t_{\frac{1}{2},p,GATA6}$ | Lifetime (or half-life) for a molecule of protein | 2 hours |
| $\tau_{p,ERK}$ | $t_{\frac{1}{2},p,ERK}$ | Lifetime (or half-life) for a molecule of protein | 48 hours |
| $\tau_{p,P-NANOG}$ | $t_{\frac{1}{2},p,P-NANOG}$ | Lifetime (or half-life) for a molecule of protein | 1 hour |
| $\tau_{p,D-FGFR-FGF4}$ | $t_{\frac{1}{2},p,D-FGFR-FGF4}$ | Lifetime (or half-life) for a molecule of protein | 240 hours |
| $k_{dime}$ | | M-FGFR-FGF4 dimerization (activation) | $10 \cdot 10^{-6}\ s^{-1}$ |
| $k_{mono}$ | | D-FGFR-FGF4 monomerization (inactivation) | $3 \cdot 10^{-3}\ s^{-1}$ |

Note. For complete details about all implemented GRN interactions related to these fixed model parameters, please check Model at cell scale. See also Table 2.

needed for reaching 50% of negative or positive gene transcriptional control; consequently, each gene-TF pair requires its own separate half-saturation threshold. This cooperativity coefficient ($k_{coop}$) is an auxiliary variable which adjusts the strength of mutual influence among TF copies; we arbitrarily defined it as $k_{coop} = 5$ to increase TF-cooperativity potency (i.e., $\uparrow k_{coop} \Rightarrow \downarrow k_u$). Specifically, we calculate the nominal promoter unbinding rates via the formula $k_u = h_{sat} k_b / k_{coop}^q$. Where, for a given gene-TF pair: $h_{sat} \in \{h_{rep}, h_{act}\}$; $k_b$ is its nominal promoter binding rate; $q$ is its maximal occupancy.

To summarize, Eq (3) illustrates the most basic reaction set of the TF promoter binding/unbinding dynamics, plus Fig 12 shows the elementary promoter architecture.

$$A + B_0 \underset{k_{u,0}}{\overset{k_{b,1}}{\rightleftharpoons}} B_1 \quad \cdots \quad A + B_{q-1} \underset{k_{u,q-1}}{\overset{k_{b,q}}{\rightleftharpoons}} B_q \tag{3}$$

Where: $A$ is a given TF; $B_Q$ is the current occupancy of a given gene promoter $B$ by $A$; $Q \in \{0, \ldots, q\}$; $q$ is the maximum number of binding sites for $A$ at $B$; $k_b$ is the nominal promoter binding rate; $k_u$ is the nominal promoter unbinding rate.

**Synthesis and degradation of mRNA.** For the transcription model, we assume that mRNA synthesis occurs as a single-step reaction but it is only possible when the gene promoter

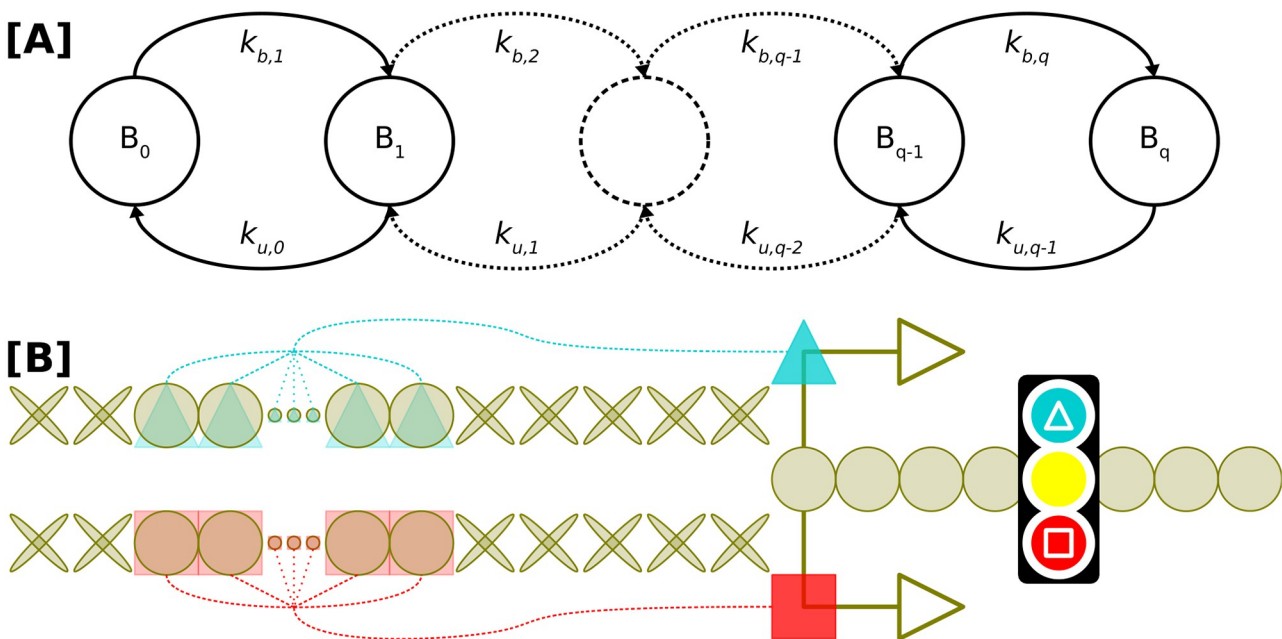

**Fig 12. Architecture of the modeled gene regulatory region.** Transcription factor (TF) *A* binds to regulatory sites *B* at an effective rate proportional to $k_b$ and unbinds from them at an effective rate proportional to $k_u$. Unlike $k_b$, $k_u$ is directly related to $q$ (which is the maximal occupancy of *B* by *A*); i.e., $k_{b,1} = \cdots = k_{b,q}$, where $q$ is the number of TF binding sites (TFBSs) for *A* at *B*. [A] Markov chain transition diagram of TF binding/unbinding dynamics. [B] Every gene has a respective regulatory region with multiple independent binding sites for each of its regulating TFs. Gene activation and repression are both cooperative: exactly $q$ TF copies must be simultaneously bound to their particular sites for activation (cyan triangles) or repression (red squares) of expression. By default, repression takes precedence over activation.

is not under control of a repressor TF; recall that we follow the "all-or-nothing" gene activation/repression configuration. We have as well accounted for two concomitant transcription modes: basal and full-induction production. Basal transcription contributes 20% of the maximal average steady-state mRNA copy number. The remaining 80% of the maximum mean steady-state mRNA copy number is contributed by full-induction transcription; which is only possible when an activator TF is occupying all of its binding sites at a given promoter. Accordingly, the mRNA synthesis rate is calculated via the formula $k_{m,s} = k_{m,s,basal} + k_{m,s,find} = \overline{M}/\tau_m$. Where: $k_{m,s,basal} = c_{basal}\overline{M}/\tau_m$; $k_{m,s,find} = c_{find}\overline{M}/\tau_m$; $c_{basal}$ and $c_{find}$ are the basal and full-induction relative contributions (i.e. $c_{basal} + c_{find} = 1$), respectively. The symbol $\overline{M} \doteq \langle M_t \rangle_{equilibrium}$ is a shorthand for the maximum mean steady-state value of mRNA copies for a given gene at full activation ($M_t$ is the number of mRNA molecules at time $t$). The symbol $\tau_m$ denotes the lifetime (or half-life $t_{\frac{1}{2},m}$) for a molecule of mRNA.

For *Nanog*, this mean mRNA value has been indicated to reach the order of hundreds of copies; approximately 100–400 molecules [72, 104, 191, 192]. For *Gata6* and *Fgf4*, there are no concrete mean mRNA values reported, but it seems they are similar to the average *Nanog*-expression level [66]. Analogously, the *Nanog*-mRNA lifetime has been reported to be around 4–5 hours [71, 103, 191, 193], the *Gata6*-mRNA lifetime has been reported to be around 3–4 hours [194, 195], and we have not found concrete reports about the *Fgf4*-mRNA lifetime.

For simplicity, we have considered the mean mRNA values for *Nanog* and *Gata6* to be the same, which classifies them as fixed parameter values (we chose $\overline{M}_{Nanog} = \overline{M}_{Gata6} = 250$ copies). In the case of *Fgf4*, its mean mRNA value is deemed to be identical to the case of the other

two genes and it is also considered a fixed parameter value. However, we additionally impose that any *Fgf4* expression must be entirely regulated by NANOG and GATA6 levels; in other words, *Fgf4* has no basal mRNA production ($\overline{M}_{Fgf4} = 200$ copies). Likewise, the mRNA half-lives for *Nanog*, *Gata6*, and *Fgf4* are determined to be the same (we chose $\tau_{m,Nanog} = \tau_{m,Gata6} = \tau_{m,Fgf4} = 4$ hours).

For the mRNA degradation mechanism, we assumed that it is a first-order process: the nominal degradation rate is simply the multiplicative inverse of the lifetime; $k_{m,d} = 1/\tau_m = \tau_m^{-1}$.

In a nutshell, Eq (4) illustrates the reaction set of the mRNA synthesis and degradation dynamics.

$$\emptyset \xrightarrow{k_{basal}} M$$
$$G \xrightarrow{k_{find}} G + M$$
$$M \xrightarrow{k_{m,d}} \emptyset$$
$$k_{basal} = \begin{cases} (1 - G_{rep})k_{m,s,basal} & \text{if} \quad G \in \{Nanog, Gata6\}, \\ 0 & \text{if} \quad G = Fgf4. \end{cases}$$
$$k_{find} = (1 - G_{rep})G_{act}k_{m,s,find}$$

(4)

Where: $G$ is a particular gene; $M$ is the mRNA of $G$; $G_{rep}$ and $G_{act}$ are indicator random variables representing the current state of $G$. $G_{rep}$ indicates a repressed gene (full promoter occupancy by repressor TF) and $G_{act}$ indicates an activated gene (full promoter occupancy by activator TF), respectively.

**Synthesis and degradation of protein.** The translation model assumes that, once mRNA is available, protein synthesis occurs as a single-step reaction. This assumption holds for NANOG, GATA6, and FGF4 proteins. For ERK, as there is no mRNA step, spontaneous activity produces its inactive protein form (I-ERK), which can undergo phosphorylation and become active (A-ERK). There are several indications for high abundance of ERK during the developing blastocyst, so it is not a limiting factor for the cell signaling process [114, 178, 196–198]. It has also been reported that ERK has a long lifetime (48–72 hours) [182, 199, 200]. To reflect this strong presence of ERK across the cellular reaction domain, every standard model simulation assigns to each cell an initial high amount of ERK. As well, ERK can be synthesized and degraded at a rate directly proportional to its chosen half-life ($\tau_{p,ERK} = 48$ hours) and its chosen maximum mean steady-state protein copy number ($\overline{P}_{ERK} = 1000$ copies). Here: $k_{p,s,ERK} = \overline{P}_{ERK}/\tau_{p,ERK}$ is the rate of ERK synthesis; $k_{p,d,ERK} = 1/\tau_{p,ERK} = \tau_{p,ERK}^{-1}$ is the rate of ERK degradation.

For the remaining protein species, NANOG-GATA6-FGF4, an extra assumption was made to incorporate an additional sense of bursty production: every mRNA molecule is capable of synthesizing on average 4 protein molecules before it decays naturally; i.e., the theoretical maximum average steady-state value of protein molecules per cell can be calculated to be $\overline{P}_{NANOG} = 1000$, $\overline{P}_{GATA6} = 1000$, and $\overline{P}_{FGF4} = 800$ copies. Bursty gene expression has been shown to significantly increase cell-cell variability of mRNA and protein levels, which itself has been suggested to enable enhanced adaptation to environmental changes and constraints [72, 201]. Thus, the nominal synthesis rates for these proteins can be calculated via the formula $k_{p,s} = \overline{P}/\tau_p$. For the cases of NANOG and GATA6, the protein half-lives have been reported to be 2–3 hours [202–204] and 1–3 hours [194, 195], respectively; we chose, for simplicity, $\tau_{p,NANOG} = 2$ hours and $\tau_{p,GATA6} = 2$ hours. Just like the previous case, the symbol

$\overline{P} \doteq \langle P_t \rangle_{equilibrium}$ is a shorthand for the maximum mean steady-state value of protein copies for a given gene at full activation ($P_t$ is the number of protein molecules at time $t$).

We assumed that any protein degradation process simply follows first-order dynamics. This condition means that, for a particular protein, its nominal degradation rate is the multiplicative inverse of its half-life: $k_{p,d} = 1/\tau_p = \tau_p^{-1}$. This assumption not only holds for these primary proteins but also applies to all the derivative molecules: P-NANOG, M-FGFR-FGF4, and D-FGFR-FGF4. For NANOG, phosphorylation by ERK reduces its stability, which in turn accelerates its degradation and essentially halves its lifetime [179]. Hence, P-NANOG half-live has been categorized as a fixed parameter value ($\tau_{p,P\text{-}NANOG} = 1$ hour). For D-FGFR-FGF4, we did not learn any concrete information about its lifetime. However, we determined that any practical decay should occur after monomerization (indirectly via transitions between dimer and monomer configurations); as such the value for D-FGFR-FGF4 half-life is set artificially high and is a fixed parameter value ($\tau_{p,D\text{-}FGFR\text{-}FGF4} = 240$ hour). The lifetime of M-FGFR-FGF4 ($\tau_{p,M\text{-}FGFR\text{-}FGF4}$) is therefore a free model parameter controlling the actual extracellular stability of FGF4-related resources. A similar challenge happens for FGF4 itself, as there are contrasting reports about its lifetime [205, 206], we made its half-life a free model parameter ($\tau_{p,FGF4}$).

In summary, maximal average steady-state protein levels for NANOG, GATA6, FGF4, and ERK are fixed model-parameter values. Protein half-lives for NANOG, GATA6, ERK, P-NANOG, and D-FGFR-FGF4 are also fixed model-parameter values. The lifetimes of FGF4 and M-FGFR-FGF4 are free model parameters. Eq (5) illustrates the reaction set of the protein synthesis and degradation dynamics.

$$M \xrightarrow{k_{p,s}} M + P \quad (P \neq ERK)$$
$$\emptyset \xrightarrow{k_{p,s,ERK}} P \quad (P = ERK) \tag{5}$$
$$P \xrightarrow{k_{p,d}} \emptyset$$

Where: $M$ is a particular mRNA; $P$ is the protein of $M$; $k_{p,s}$ and $k_{p,d}$ are the protein synthesis and degradation rates, respectively.

**FGFR activation and inactivation.**   Multiple experimental studies have demonstrated that FGF signaling is a fundamental coordinator of the ICM specification into EPI and PRE populations [44, 67, 177, 180, 181]. They have also indicated that FGF4 binds to two receptors: FGFR1 and FGFR2. However, FGFR1 plays the main role in the ICM cell-fate establishment and FGFR2 has a supporting/redundant character in the PRE-lineage regulation [67, 177]. Additionally, FGFR1 is expressed abundantly all over the ICM [140], plus FGF4 signaling via FGFR1 is critical for maintaining physiological levels of NANOG in EPI cells to help them reach primed pluripotency [67]. Nonetheless, we do not model explicitly any FGF receptor; instead, for our model simulations, once FGF4 is available at the plasma membrane, we simply consider it to be the receptor-ligand complex in its inactive (monomer) form. In other words, whenever FGF4 undergoes a diffusive-jump event we relabel it as the monomer complex M-FGFR-FGF4. Furthermore, FGFR activation requires ligand-receptor dimer assembly, which in turn makes possible biochemical transduction of FGF signaling [140, 207]. As such, the D-FGFR-FGF4 (dimer) complex represents this active form triggering the FGF/ERK pathway in our system model.

The nominal rates of FGFR dimerization (activation) and monomerization (inactivation) are fixed model-parameter values. For the dimerization case, we follow the theory of diffusion-controlled reactions in a similar manner to the gene promoter scenario [185]: we use the same formula as for $k_b$ but we assume a receptor-ligand complex diffusion constant 30 times slower than the typical TF diffusion constant. For the monomerization case, we assume that FGFR1

phosphorylation and dephosphorylation follow similar kinetics, thus we take the average time (approximately 360 seconds) for reaching the half-saturation point based on some experimental receptor-ligand (FGFR-FGF4) response curves [140, 207].

In brief, Eq (6) recaps the reaction set of FGFR dimerization and monomerization.

$$M\text{-}FGFR\text{-}FGF4 \quad + \quad M\text{-}FGFR\text{-}FGF4 \quad \underset{k_{mono}}{\overset{k_{dime}}{\rightleftharpoons}} \quad D\text{-}FGFR\text{-}FGF4 \tag{6}$$

Where: $k_{dime}$ and $k_{mono}$ are both fixed parameter values; $k_{dime} = 4\pi d(D/30)/V$; $k_{mono} = 1/360$ s$^{-1}$.

**ERK activation and inactivation.** Proper stimulation of the FGF/ERK signalling pathway is a requisite for ICM cell differentiation during mouse blastocyst formation [208, 209]. This stimulation is also essential for escaping naive pluripotency in mouse embryonic stem cells (ESCs) [114, 208]. It has been shown experimentally that activation of ERK occurs via phosphorylation; this mechanism is present at ICM progenitors as well as PRE and EPI tissues [209]. After all, ERK is the main FGF-signaling effector, which relays FGF4 fluctuations downstream of FGFR1 and FGFR2. This signaling pathway is a basic component for the regulation of cellular differentiation and homeostasis [200].

Previous reports have suggested that ERK experiences highly heterogeneous dynamics [128, 178]. This high variability has been regarded as an additional layer of plasticity, which could enhance the cell-fate decision process by augmenting cellular heterogeneity during ICM/blastocyst development [7]. As such, ERK activity has been extensively examined in recent studies [7, 114, 128, 178]. Nevertheless, quantitative information about important reaction rates, relevant molecular concentrations, and many other kinetic parameters remains mostly elusive. Here, we use current in vitro and in silico reports on oscillations of ERK nuclear translocation for related biological systems, as well as experimental descriptions of ERK activity dynamics when exogenous FGF4 is present in mouse ESC systems [9, 196]. We gathered all this information in order to establish generous bounds for the free model parameters representing the ERK phosphorylation (activation) and dephosphorylation (inactivation) reaction kinetic rates.

Fortunately, there are several succinct studies indicating that ERK has a long lifetime (48–72 hours) [199, 200]. They also report that there is no concrete evidence for feedback mechanisms (or stimulus-induced changes) regulating its protein expression, plus they indicate that ERK displays high physiological protein levels [198]. These observations allows us to treat the collection of ERK molecules as a pool of ready-to-use kinases; in other words, the availability of ERK is not a limiting factor for cell-cell communication, rather its influence on the ICM specification process is controlled upstream of the release of FGF4 into the extracellular environment. Therefore, there is a need for tuning the reaction rates of ERK activation and deactivation.

To recap: we chose $\tau_{p,ERK} = \tau_{p,I\text{-}ERK} = \tau_{p,A\text{-}ERK} = 48$ hours; $k_{pho,A\text{-}ERK}$ (phosphorylation) and $k_{doh,I\text{-}ERK}$ (dephosphorylation) are free parameters. Eq (7) encapsulates the ERK activation and inactivation reaction dynamics.

$$
\begin{aligned}
D\text{-}FGFR\text{-}FGF4 \quad + \quad I\text{-}ERK \quad &\xrightarrow{k_{pho,A\text{-}ERK}} \quad D\text{-}FGFR\text{-}FGF4 \quad + \quad A\text{-}ERK \\
A\text{-}ERK \quad &\xrightarrow{k_{doh,I\text{-}ERK}} \quad I\text{-}ERK
\end{aligned}
\tag{7}
$$

**NANOG phosphorylation and dephosphorylation.** The *Nanog* gene plays a key role in the regulation of pluripotency, self-renewal, and differentiation potential in human/mouse

ESCs as well as early-embryo development [105, 113, 179]. Thus, tight control of *Nanog* expression is of relevance for the correct progression of such developmental systems. This control is partially executed by ERK via two complementary mechanisms working at distinct scales. Firstly, A-ERK can recruit other proteins to the *Nanog* locus and repress its transcriptional activity [105]. Along with this inhibition of *Nanog* transcription, A-ERK phosphorylates NANOG, which directly leads to a reduction of its protein stability and an increase of its degradation rate [105, 179]. All together, these observations suggest that this NANOG-regulation motif is indirectly orchestrated by NANOG itself. In this sense, ERK-mediated control of NANOG protein levels can be seen as a negative autocrine feedback loop (indirect autorepression), which might emerge following high FGF signaling. As we mentioned before, NANOG phosphorylation by A-ERK decreases/halves its lifetime, which in consequence implicitly decreases/halves its average steady-state protein copy number ($\overline{P}_{P-NANOG} \approx \overline{P}_{NANOG}/2$) [113]. However, we do not have access to any other concrete data about the kinetic parameters related to this process. For this reason, the transition rates between NANOG and P-NANOG must enter our model as inferable parameters.

In short, $\tau_{p,P\text{-}NANOG}$ = 1 hour is a fixed value, while $k_{pho,P\text{-}NANOG}$ and $k_{doh,NANOG}$ are both free parameters. Eq (8) portrays the NANOG phosphorylation and dephosphorylation reaction set.

$$
\begin{aligned}
A\text{-}ERK \quad + \quad NANOG \quad &\xrightarrow{k_{pho,P\text{-}NANOG}} \quad A\text{-}ERK \quad + \quad P\text{-}NANOG \\
P\text{-}NANOG \quad &\xrightarrow{k_{doh,NANOG}} \quad NANOG
\end{aligned}
\tag{8}
$$

**Model at tissue scale.** Technically, all events (reactions or diffusive jumps) share the same essential characteristics under the SSA formulation. However, we make an explicit distinction between reaction events and jump-diffuse events for two reasons: first, it is a convenient abstraction for distinguishing multiple temporal and spatial scales; second, their respective rates are calculated or estimated in contrasting manners because they depend on distinct features of our system model. Consequently, we have separated the focal bulk of the signaling model from the other reaction groups, and we treat it as a cohesive submodel at tissue level.

The most fundamental idea surrounding our signaling-model approach concerns the concept of conservation of molecular flux. In simple words, once the signaling molecule undergoes a jump-diffusion event, the probability of arrival over each of the cell neighbors (which includes the origin cell itself) must be a conserved feature. This conserved feature depends on the neighborhood configuration and has to be calculated for each cell. For our system model, the full ICM neighborhood representation is a rectangular voxel grid with a one-cell thickness, which mimics a monolayer or 2D cellular culture; each voxel emulates an embryonic cell.

In this sense, two cells (voxels) are categorized as first-degree neighbors only when they share a complete face; they can communicate directly with each other. In other words, if they only share a single edge (2 adjacent vertices), then they are not categorized as first-degree neighbors (any communication occurs indirectly between them); however, they are categorized as second-degree neighbors. This requisite implies that a particular cell must have strictly 2, 3, or 4 members within its first-degree neighborhood (which does not include itself). Thus, by recursion, it is easy to construct high-order neighborhoods; analogously, it is easy to identify low-degree neighborhood decompositions. By definition, each cell by itself forms a zero-degree neighborhood.

We model the release of FGF4 as a jump-diffuse event. When FGF4 molecules experience diffusive jumps, they cooperatively act as a biochemical signal, which is transduced by the respective FGFRs into an intracellular response involving multifold messenger molecules. This

signal ultimately stimulates gene expression changes which consequently promote cellular adaptability [139]. For our system model, FGF4 signaling manifests via autocrine and paracrine feedback loops. We also have included, for completeness, a third FGF4-signaling mode: membrane-level exchange of ligand molecules. Although, to the best of our knowledge, there is no experimental study of this auxiliary signaling mode yet, it is natural to think in terms of our modeling framework that once a molecule of FGF4 is available at the cellular membrane, it can experience ligand internalization together with its receptor, or it can sustain diffusive jumps between neighboring membranes. Accordingly, these two premises enter our final reaction set.

We assume that there is a primary flux transporting/releasing FGF4 molecules into the extracellular domain. For our implementation, this molecular flux is modeled as a first-degree process whose reaction rate depends on factors such as cellular geometry, spatial distribution of molecular escape channels, intracellular diffusion constant of FGF4, and many other features. We decided to treat this transport/release rate as a free model parameter because we only have access to some well-informed estimates for its bounds. These bounds are based on several studies of first-passage time distributions for related theoretical problems [210, 211].

Subsequently, the FGF4 stream is split into two secondary fluxes. This split is controlled through the free-parameter relationship $\chi_{auto} + \chi_{para} = 1$, which dictates the probability ratio of ligand binding to its origin cell ($\chi_{auto}$) or one of its neighboring cells ($\chi_{para}$). In short, the exit rate of FGF4 molecules ($\chi_{escape}$) eventually gives raise to two complementary signaling channels: autocrine and paracrine loops.

In a similar fashion, the exchange rate of FGF4 between cell membranes ($k_{exchange}$) is assumed to be limited by ligand-receptor affinity, just like for the case of FGFR monomerization [207]. As such, generous bounds for this rate were placed and its concrete value is treated as a free parameter.

**Autocrine signaling.** The autocrine feedback loop is thought to play an essential role for the *Nanog* self-regulation. In that regard, the collateral *Nanog* auto-repression is a self-perpetuating process maintaining physiologically-relevant NANOG levels. This process is fundamental because it allows an EPI cell to enter a state of primed pluripotency which supports the correct developmental progression [105]. Within our model, the rate of autocrine signaling is denoted by $k_{escape,auto}$, and it is calculated via the formula $k_{escape,auto} = \chi_{auto}\rho_{auto}k_{escape}$. Here, the argument $\rho_{auto} = (1 - \chi_{para}\rho_{meme})/\rho_{auto}$ involves the variable $\rho_{meme}$ which quantifies the fraction of surface area shared between a particular cell and its first-degree neighborhood.

**Paracrine signaling.** The paracrine feedback loop is deemed to have a critical role in inducing the PRE fate. The dose-dependent upregulation of *Fgf4* by NANOG prompts FGF4 paracrine communication, which triggers a reaction cascade concurrently downregulating NANOG and upregulating GATA6 levels [102, 105]. Within our model, the rate of paracrine signaling is denoted by $k_{escape,para}$, and it is calculated via the formula $k_{escape,para} = \chi_{para}\rho_{meme}k_{escape}$.

**Membrane exchange.** We denote the exchange rate of FGF4 molecules between cellular membranes as $k_{exchange}$. This FGF4 exchange rate is independent of the primary FGF4 secretion rate ($k_{escape}$); however, it must be adjusted relative to the actual number of first-degree neighbors for each particular cell. Thus, it is preferable to think of $k_{exchange}$ as the maximal membrane-level FG4-exchange rate, which is only possible when a given cell completely shares all of its faces with other neighbors ($\rho_{meme} = 1$). In other words, we assume that the practical FGF4-exchange rate $k_{meme}$ is simply directly proportional to the maximum FGF4-exchange rate $k_{exchange}$, where $0 \leq k_{meme} \leq k_{exchange}$. This constraint enters our model via the formula $k_{meme} = \rho_{meme}k_{exchange}$.

To finalize, Eq (9) illustrates the reaction set of FGF4 autocrine, paracrine, and membrane-exchange communication modes.

$$FGF4 \xrightarrow{k_{escape,auto}} M\text{-}FGFR\text{-}FGF4 \quad \text{(Self-Cell)}$$

$$FGF4 \xrightarrow{k_{escape,para}} M\text{-}FGFR\text{-}FGF4 \quad \text{(Other-Cell)} \tag{9}$$

$$\text{(Self-Cell)} \quad M\text{-}FGFR\text{-}FGF4 \xrightarrow{k_{meme}} \text{(Other-Cell)} \quad M\text{-}FGFR\text{-}FGF4$$

**Inventory of model parameter values.** We provide here two complementary tables summarizing the most significant model parameters presented so far. Table 3 recaps compactly all the fixed values we have either chosen based on well-informed estimates or taken from our literature sources. Table 4 presents a quick view of the model parameters we have categorized as free values and their respective ranges. These parameter ranges were derived from data found across all our literature sources, and they generally represent educated guesses made by collecting information about closely related biological systems. Nonetheless, we largely assumed generous bounds for all these ranges, delegating the search for biophysically-relevant values to our parameter inference scheme.

**Table 4. Summary of inferred (free) model parameters.**

| Name | Alias | Description | Lower Bound | Inferred Value | Upper Bound | Units |
|---|---|---|---|---|---|---|
| $h_{\text{act},Nanog,\text{NANOG}}$ | Nanog_NANOG | Half-saturation level (self-activation) | 0 | 125 | 1000 | [pc] |
| $h_{\text{act},Gata6,\text{GATA6}}$ | Gata6_GATA6 | Half-saturation level (self-activation) | 0 | 275 | 1000 | [pc] |
| $h_{\text{rep},Gata6,\text{NANOG}}$ | Gata6_NANOG | Half-saturation level (mutual-repression) | 0 | 412 | 1000 | [pc] |
| $h_{\text{rep},Nanog,\text{GATA6}}$ | Nanog_GATA6 | Half-saturation level (mutual-repression) | 0 | 411 | 1000 | [pc] |
| $h_{\text{act},Fgf4,\text{NANOG}}$ | Fgf4_NANOG | Half-saturation level (activation) | 0 | 552 | 1000 | [pc] |
| $h_{\text{act},Gata6,\text{A-ERK}}$ | Gata6_A-ERK | Half-saturation level (activation) | 0 | 615 | 1000 | [pc] |
| $h_{\text{rep},Fgf4,\text{GATA6}}$ | Fgf4_GATA6 | Half-saturation level (repression) | 0 | 95 | 1000 | [pc] |
| $h_{\text{rep},Nanog,\text{A-ERK}}$ | Nanog_A-ERK | Half-saturation level (repression) | 0 | 695 | 1000 | [pc] |
| $\tau_{\text{escape}}$ | $k_{\text{escape}}^{-1}$ | Mean escape time (FGF4) | 300 | 2303 | 4500 | [s] |
| $\tau_{\text{exchange}}$ | $k_{\text{exchange}}^{-1}$ | Mean exchange time (FGF4) | 30 | 1380 | 4200 | [s] |
| $\chi_{\text{auto}}$ | $1 - \chi_{\text{para}}$ | Autocrine signaling fraction | 0 | 0.39 | 1 | |
| $\tau_{\text{pho,ERK}}$ | $k_{\text{pho,A-ERK}}^{-1}$ | Half-turnover time (phosphorylation) | 300 | 31379 | 43200 | [s] |
| $\tau_{\text{doh,ERK}}$ | $k_{\text{doh,I-ERK}}^{-1}$ | Half-turnover time (dephosphorylation) | 30 | 1025 | 43200 | [s] |
| $\tau_{\text{pho,NANOG}}$ | $k_{\text{pho,P-NANOG}}^{-1}$ | Half-turnover time (phosphorylation) | 300 | 18944 | 43200 | [s] |
| $\tau_{\text{doh,NANOG}}$ | $k_{\text{doh,NANOG}}^{-1}$ | Half-turnover time (dephosphorylation) | 30 | 21361 | 43200 | [s] |
| Mean Initial mRNA Count | Nanog_Gata6 | Initial condition | 0 | 117 | 250 | [mc] |
| Mean Initial PROTEIN Count | NANOG_GATA6 | Initial condition | 0 | 482 | 1000 | [pc] |
| $\tau_{\text{d,FGF4}}$ | $k_{\text{p,d,FGF4}}^{-1}$ | Lifetime or half-life | 300 | 13158 | 28800 | [s] |
| $\tau_{\text{d,M-FGFR-FGF4}}$ | $k_{\text{p,d,M-FGFR-FGF4}}^{-1}$ | Lifetime or half-life | 300 | 3456 | 28800 | [s] |

Note. For complete details about all implemented GRN interactions related to these inferred (free) model parameters, please see Model at cell scale and Model at tissue scale. The two parameters "Mean Initial mRNA Count" and "Mean Initial PROTEIN Count" have not been properly introduced yet, but their usage will be explained at Model parameter inference framework. Notation: act = activation; rep = repression; pho = phosphorylation; doh = dephosphorylation; [pc] = [protein copies]; [mc] = [mRNA copies]; [s] = [seconds]. See also S1 and S2 Figs.

### Model parameter inference framework

The exploration of the immense parameter space of a mechanistic model is a demanding computational task, especially for biophysical problems dealing with spatial-stochastic system representations [135]. Several classical inference/optimization methods such as heuristic tuning, stochastic gradient descent, simulated annealing, and approximate Bayesian computation (ABC) have been traditionally used for finding reasonable parameter sets of biophysical mechanistic models [89, 95–97, 212]. Nonetheless, all of these aforementioned methods are often computationally inefficient because they require many expensive simulations for properly scanning the model parameter space, and they commonly lack powerful parameter-space interpolation strategies. As a consequence, these inference methods become prohibitively ineffective when they are applied to high-dimensional spatial-stochastic models, which are nowadays usually employed to represent complex biological systems.

Here, by leveraging our access to high-performance computing (HPC) resources, we take advantage of the sequential neural posterior estimation (SNPE) algorithm and combine it with classical ML concepts, in order to implement a comprehensive model parameter inference framework. The SNPE algorithm is part of a ground-breaking family of likelihood-free inference techniques [86, 87, 89]. These state-of-art simulation-based inference (SBI) algorithms fundamentally rely on artificial neural networks (ANNs) to approximate model parameter probability distributions; the respective ANNs are trained with simulation/synthetic data, but are conditioned on target/experimental observations [89, 95].

In our case, this original framework was applied to infer two individual parameter sets for two separate models, which are capable of recapitulating the most fundamental characteristics of the fully-formed mouse blastocyst. The first model represents the wild-type variant of the underlying biophysical system, and it is our principal system model; in other relevant contexts, we also refer to it as the "inferred-theoretical wild-type" or ITWT. This system has a functional cell-cell communication. The second model is an auxiliary system; all the parameter values of the ITWT are reused for this supplementary system, except for the core GRN components which are reinferred in a loss-of-function mutant setting: this system model has a nonfunctional cell-cell communication. We consequently refer to it as the "reinferred-theoretical mutant" or RTM.

To illustrate the key stages of our exploratory Bayesian inferential framework, we present a concise list of the most important ideas of our workflow. For additional information concerning some general considerations of our model parameter inference framework, please see Fig 1 and S1 Appendix.

**Constructing the prior distribution.** A key stage of our workflow is the construction of the model parameter prior distribution. Once every suitable model parameter has an appropriate fixed value (see Tables 2 and 3), it is necessary to condense every belief/intuition about the remaining free model parameters into a reasonable prior distribution. In this work, this prior is a multivariate uniform distribution with 19 dimensions (vector components). Every such vector component has a predefined range; those ranges are derived from educated guesses informed by data on closely related biological systems or found across all our literature sources. We largely assume rich but realistic bounds for all of those ranges (see Table 4), as ultimately the aim is to delegate the search for biophysically-relevant values to our parameter inference scheme. But while the general idea applies to both system models (ITWT and RTM), the RTM prior distribution has only 4 dimensions corresponding to the core GRN motif.

Two of the freely-varying model parameters fulfill a particular role, namely "Mean Initial mRNA Count" and "Mean Initial PROTEIN Count". These parameters are used to define meaningful initial condition distributions (ICDs), as there is no detailed experimental

information currently available about them. In the typical setting of our simulation procedure, they operate together as the mean-value vector of an ICD which is itself a composition of Poissonian and binomial distributions (see Computational experiments); as such, they also dictate the variance matrix of this ICD. From that ICD, we sample the starting *Nanog-Gata6* mRNA and NANOG-GATA6 (protein) copy numbers; accordingly, this sampling is performed per simulated cell. In this regard, the only stipulated hard molecular constraint follows from imposing the maximal mean mRNA and protein copy numbers reached at full induction: 250 and 1000 molecules, respectively.

**Simulation data generation.**    At this stage, we explicitly incorporate two of the basic empirical observations we are aiming at recapitulating into the simulation procedure itself. Highly reproducible ratio of 2: 3 for EPI-PRE lineages [38, 99]; absence of FGF4-mediated signaling (no spatial coupling) forces the ICM to almost exclusively adopt the EPI fate [21, 73]. See also S1 Appendix. Via parallel computing, our simulator generates after each run a composite trajectory for two complementary configurations of our system model: functioning cell signaling, and nonfunctioning cell signaling. When cell signaling is functioning, the targeted proportions are 40% for the EPI population and 60% for the PRE population. Whereas, when cell signaling is nonfunctioning, the targeted proportions are 100% for the EPI population and 0% for the PRE population. While these two configurations are necessary for correctly deriving the ITWT system, the valid derivation of the RTM system requires only one configuration: as cell-cell communication is always turned off for the RTM, here the targeted proportions are 40% for the EPI lineage and 60% for the PRE lineage.

Furthermore, each run should produce at least 48 hours of simulated time to sufficiently capture the pertinent model dynamics; for additional information, see Key stages of mouse embryo preimplantation development. By design, the simulation data generation can be executed independently of any other stage, which allows the production of a sufficiently large trajectory batch per run. This simulation batch size is arbitrary and totally influenced by the available computational resources; in our case, the simulation batch size is generally 100 thousand (composite) trajectories per each ANN training round.

**Training data generation.**    The raw simulation data is ineffective for training the ANN, because of the high dimensionality and the multiscale character of the generated trajectories. Via sequential data transformations, we create features that can be used for successfully training the ANN. In practical terms, these features are low-dimensional projections of high-dimensional temporal-spatial stochastic dynamics data.

Our simulations are genuinely event-driven and therefore produce trajectories irregularly spaced in time. As a preparative step, we accordingly first resample the simulated data onto a regular time grid. For simplicity, we always generate a resampled time series with a sampling period equal to 0.25 hours (or 15 minutes); we deemed that sampling period to be sufficient for fully characterizing the underlying temporal dynamics on the tissue scale.

All subsequent steps are performed on that regularized time series. Per simulated cell, the resampled time series represents the dynamics of all the involved biochemical species. In order to avoid tracking all of the many biochemical species in our model, we focus on three system observables that characterize the specification process at cell scale: total NANOG, which is the aggregate sum of the available unphosphorylated NANOG and phosphorylated NANOG protein molecular counts; total GATA6, which is simply the GATA6 protein molecular count; total FGF4, which is the aggregate sum of all the available FGF4 proteins at both cytoplasm and membrane levels (including receptor-bound FGF4 molecules). While total FGF4 is only useful for analyzing our system under perturbed conditions, total NANOG and total GATA6 are the guiding drivers of our training data generation: these are the main markers

determining the lineage for every cell at each simulated time point; for additional details, check Computational experiments.

We next further group the cell-level observables into a tissue-level one. The key tissue-scale observable variable is the total count for each of the three possible cell fates at each simulated time point. As such, this tissue-level observable is used to define a "pattern score" that meaningfully discriminates the targeted/idealized patterning behavior from undesired patterning behaviors of our system. This constitutes the most important part of our data-transformation pipeline, and it is fully described in the next part.

**Constructing the pattern score (objective) function.**   To generate the actual ANN training dataset, two closely connected steps are required for bringing it to fruition. First, at every time point for each possible fate, the total cell count is compared against the target cell count. Moreover, there is a specific target cell count per each prescribed configuration of the respective system: for the ITWT, the EPI-PRE-UND lineage target proportions are 40-60-0 percent with functioning spatial coupling, and 100-0-0 percent when spatial coupling is inactivated; the RTM (in which spatial coupling is always inactivated) has the EPI-PRE-UND lineage target proportions of 40-60-0 percent. The resultant is a set of marginal scores, one per system configuration; i.e., two marginal scores for the ITWT, and one for the RTM. Second, all these resulting (marginal) configuration scores must be combined into a discriminatory pattern score, which measures the distance between a particular simulation score and the idealized behavior score.

The (joint) pattern score time series performs as the outcome of a vector-valued objective function whose main inputs are the total cell count and the target cell count per system configuration, for each possible fate at every time point. Therefore, our essential goal is to intelligently optimize (maximize) this objective function. As such, the joint pattern score is the foundational element of the ANN training dataset.

For the first step, in formal terms, let $\mathbf{Z}_t = (Z_{t,0}, Z_{t,1}, Z_{t,2})$ be a discrete random vector taking values in $\mathbb{N}^3$ (naturals or nonnegative integers), which represents the total cell count for each lineage at a given discrete time point $t \in \mathbb{N}$. For simplicity, the three possible cell lineages are arbitrarily indexed as 0 (EPI), 1 (PRE), and 2 (UND). Also, let $\mathbf{w}_m = (w_{m,0}, w_{m,1}, w_{m,2}) \in \mathbb{N}^3$ be a discrete vector representing the target cell count for each lineage, and for a given system configuration which is arbitrarily indexed by $m \in \{0, 1\}$. The configuration score is thus the continuous random variable $S_{t,m}$ taking values in $\mathbb{R}$ (reals), which is a nonlinear transformation mapping from an absolute-difference vector $|\mathbf{z}_t - \mathbf{w}_m| = (|z_{t,0} - w_{m,0}|, |z_{t,1} - w_{m,1}|, |z_{t,2} - w_{m,2}|)$ to a point $s_{t,m}$ in the closed interval $[0, 1]$; i.e.:

$$S_{t,m} = \frac{\exp(-\|\mathbf{Z}_t - \mathbf{w}_m\|_1/\|\mathbf{w}_m\|_1) - \overset{\star}{w}}{1 - \overset{\star}{w}} \qquad (10)$$

Here $\star w = \exp(-2\max(\mathbf{w}_m)/\|\mathbf{w}_m\|_1)$, $\|\mathbf{w}_m\|_1/\|\mathbf{Z}_t\|_1 = 1$, and the expression $\|\cdot\|_1$ indicates the $\ell^1$ vector norm.

For the second step, in formal terms, let $S_{t,0}$ and $S_{t,1}$ be the marginal scores for the prescribed configurations of the ITWT; clearly, this step is not applicable to the RTM, as it has only one configuration. While this step is far from being a trivial process, to combine these two marginal scores into one pattern score, we employed an $\ell^1$-inspired penalty method. This penalty directly affects the sum of the configuration scores, and it is intended to increase the discriminatory power of the joint score by favouring similarly high marginal scores, as

described by the following formula:

$$S_t = \frac{(S_{t,0} + S_{t,1}) - \overset{\star}{S}_t}{2} \tag{11}$$

Here, $\star S_t = |S_{t,0} - S_{t,1}|$ is the penalty term.

By applying $S_t$, we construct a time series spanning the last 12-hour window of the predetermined simulation period (from 0 hours to 48 hours), and by producing many such time series, we generate the effective ANN training dataset.

**Training the ANN.** The key idea of training an ANN is to use a relatively small number of simulations. The exact number of trajectories generated by the simulator should be determined by aiming at meaningfully balancing both the algorithm's computational feasibility and the ANN training reliability. This balance enables an effective exploration of the massive parameter space for the proposed system model via the trained ANN [135].

To this end, we exploited a state-of-art SBI toolbox which allowed us to easily integrate these novel AI-powered algorithms into our workflow [137]. Specifically, we employed the SNPE procedure to train a deep neural density estimator which directly estimates the model parameter posterior distribution conditional on a goal observation. As our training dataset is already a convenient latent representation of the simulation data, this stage was relatively straightforward. Furthermore, there was no need to tune the algorithm hyperparameters; the applied SBI toolbox has reasonable preset values for them.

Altogether, we trained ANNs to successfully predict model parameter sets capable of recapitulating the targeted patterning behaviors of the systems under study.

**Constructing a synthetic target/goal observation.** At heart, our data-transformation pipeline generates a suitable latent (feature) space representation of the simulated dataset for training the ANN, which strongly facilitates predicting inferred parameter distributions that agree with a prescribed effective behavior in that latent space.

Ultimately, our (synthetic) goal observation is simply a score time series whose support spans the last 12-hour window of the prescribed simulation period; in agreement with one of the basic empirical observations: EPI and PRE populations should reach their expected fate proportions roughly 8 or 12 hours before the end of the preimplantation period [2, 3, 30, 37]. See also S1 Appendix. Moreover, each entry/value of this time series is a "1": the ideal score of the target patterning behavior.

**Selecting a posterior distribution.** To account for the inherent stochasticity of the ANN training algorithm (mini-batch stochastic gradient descent) and the implicit randomness of the pattern score trajectories, we train multiple ANNs with the same dataset. Each trained ANN produces a posterior distribution, and from it we compute the maximum-a-posteriori (MAP) estimate of all the model parameters. Using each distinct MAP estimate, we generate an extra simulation batch with many fresh pattern score time series. Moreover, we construct a pattern score distribution for each of these extra batches. The idea is not only to maximize the accuracy of the underlying system behavior, but also to increase its precision, optimizing its robustness. As such, we measure the quality of each separate MAP estimate and summarize it into a single number, in order to select the best learned posterior distribution of the model parameters conditional on the target observation.

This quality measure of the MAP estimate is described by Eq (12), and it is accordingly referred to as the "meta score" $\overline{S}$ (see also Fig 1).

$$\overline{S} = \text{mean}(\max(\boldsymbol{\alpha}_{50} - \boldsymbol{\beta}, \vec{0})) \tag{12}$$

Where $\overline{s} \in [0, 1] \subset \mathbb{R}$ ($\overline{s}$ is the realization of the random variable $\overline{S}$), $\alpha_{50}$ denotes the

50-percentile (or median) vector of a given pattern score distribution at each simulated time point, $\boldsymbol{\beta} = ((\boldsymbol{\alpha}_{95} - \boldsymbol{\alpha}_5)/2)^2$ is an element-wise penalty vector favouring high accuracy together with high precision, $\vec{0}$ denotes the corresponding zero vector, and the maximum operator is applied to each component of the input vector pair. Thus, the selected model parameter set has the highest associated (MAP estimate) meta score among the available ones.

**Performing multiple rounds of inference.** In general terms, single-round inference is not enough for learning truly-applicable model parameter values, even with a significantly high simulation budget. The problem of amortized inference is the wasteful use of the simulated trajectories: the ANN effectively estimates the posterior distribution for all the possible goal observations across the entire prior space of the model parameters.

If there is only one prominent target observation (just like our case), then it is advantageous to perform multiple sequential rounds of (non-amortized) inference. The model parameter search will be focused on this single target observation. This process can be performed arbitrarily many times until the current-round score time series matches as close as possible the target score time series. In this study, we performed 8 consecutive rounds of inference producing 800 thousand (composite) simulations in total for the ITWT system, and we performed 4 consecutive rounds of inference producing 400 thousand simulations in total for the RTM system.

In this context, it is also worth mentioning that the next-round prior does not necessarily need to be the current-round posterior. The proposal distribution can be iteratively adapted to fully leverage the power of multi-round inference, which is all based on the earliest prior and the latest posterior; as such, it is possible to obtain a well-informed next-round mixture distribution [87]. While we do not exploit this technique, it can be easily integrated into our workflow.

## Computational experiments

The initial gene expression profile per cell is commonly sampled from a composition of two distributions: the (first) Poissonian part whose mean-value vector construction is dictated by the two parameters "Mean Initial mRNA Count" and "Mean Initial PROTEIN Count"; the (second) binomial part which fairly splits the respective cellular resources. While it is easy to change the goal biochemical species, we generally target only the ICDs of the two main players: *Nanog-Gata6* mRNA and NANOG-GATA6 (protein) copy numbers. As such, every typical simulation starts with reasonably balanced cell resources, which guarantees the UND state across the tissue despite the inherent randomness of the initial conditions.

For the starting test of robustness to initial condition perturbations (ICPs), one extra layer of variability is necessary. Instead of using only a two-element composition, the original ICD now integrates an additional layer which performs uniform sampling of mRNA and protein copy numbers per cell. This uniform perturbation of cellular resources is thus the ICD root component for this testing case, whereas the original layers are the other two remaining elements. Simply put, firstly a random number is uniformly sampled from a particular discrete range ([0, 250] for mRNA and [0, 1000] for protein species), secondly this preceding number is employed as the mean value of a Poisson distribution which is sampled accordingly, and thirdly the eventual molecular count for a particular biochemical species is sampled from a fair binomial distribution whose (independent) trial-number parameter is dictated by the precursory Poissonian layer. It is hence easy to see that this ICP greatly increases the early variance of the goal cell resources.

The cell-fate classification thresholds were purposefully chosen to ease the ICM cellular categorization based on a proper evaluation of the most important GRN relationships. This cell-

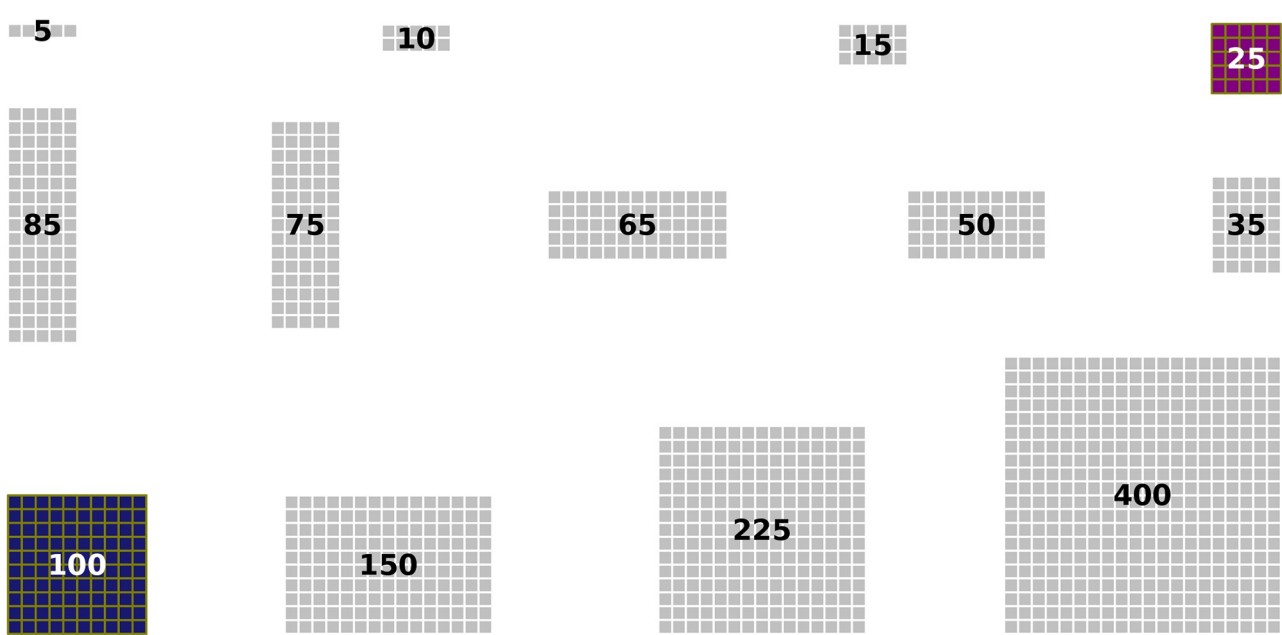

**Fig 13. Overview of cell-grid sizes employed for inference and simulations.** The 25-cell grid size (purple highlighting) was used invariably for the actual model parameter inference scheme. All test simulations (assessment of system properties such as robustness in wild-type-like and mutated conditions) employed the 100-cell tissue size (dark-blue highlighting). The remaining cell-grid sizes (gray) were solely used to study tissue-level noise (see also Fig 5).

lineage categorization is performed by comparing NANOG and GATA6 (protein) levels against constant threshold levels. Assuming Poissonian noise, a low NANOG/GATA6 cell classification occurs when the respective protein level is below a threshold of approximately 329 copies (mean basal protein level plus five times its standard deviation), a high NANOG cell classification occurs when the respective protein level is above a threshold of approximately 388 copies (mean full-induction-phosphorylation protein level minus five times its standard deviation), and a high GATA6 cell classification occurs when the respective protein level is above a threshold of approximately 842 copies (mean full-induction protein level minus five times its standard deviation). Thus, the EPI categorization occurs when a cell simultaneously displays the low-GATA6 alongside high-NANOG states, the PRE classification occurs when a cell simultaneously displays the low-NANOG alongside high-GATA6 states, and the UND categorization occurs when a cell displays any other combination of states.

For the actual model parameter inference procedure, a 25-cell grid size was employed consistently. The 100-cell tissue size was used for all the test simulations. Any other cell-grid size was exclusively employed to study noise at tissue level (Fig 5). To see a graphical comparison of every tissue size, please check Fig 13.

**Computational simulation details.** The simulator was fully implemented in the Python programming language. There are multiple such software packages supporting the simulator design: NumPy, Numba, SciPy, PyTorch, SBI, Matplotlib, Seaborn, among many others. All the computational simulations were performed in the general-purpose HPC cluster Goethe-HLR belonging to the Goethe-Universität Frankfurt am Main.

## Supporting information

**S1 Appendix. AI-powered model parameter inference: analysis, considerations, and future directions.** Summary of inferred model parameter interactions (posterior distribution).

General considerations for model parameter inference. Reflection and outlook.
(PDF)

**S1 Fig. Summary of inferred core GRN motif interactions (ITWT versus RTM).** The central component of our inference scheme is the sequential neural posterior estimation (SNPE) algorithm. Both unconditional (top row [A, B]) and conditional (bottom row [C, D]) posterior parameter distributions were obtained following 8 consecutive rounds of inference. 800 thousand composite simulations were performed for the inferred-theoretical wild-type (ITWT) system (left column [A, C]). For the reinferred-theoretical mutant (RTM) system (right column [B, D]), 4 consecutive rounds of inference were performed producing 400 thousand simulations. For complete details of the model parameter inference procedure, see Model parameter inference framework. **[A, B]** Model parameter posterior distribution. For ease of visualization, we only show the one-dimensional projection of all posterior components representing the core GRN motif interactions. **[C, D]** First assessment of model parameter sensitivity. These panels show the same components as in [A, B] but the posterior is now conditioned on the maximum-a-posteriori probability (MAP) estimate of the model parameters.
(PDF)

**S2 Fig. Summary of inferred signaling and other model parameter interactions (ITWT only).** The central component of our inference scheme is the sequential neural posterior estimation (SNPE) algorithm. Both unconditional (top rows [A-D]) and conditional (bottom rows [E-H]) posterior parameter distributions were obtained following 8 consecutive rounds of inference. 800 thousand composite simulations were performed for the ITWT system. For complete details of the model parameter inference procedure, see Model parameter inference framework. **[A-D]** Model parameter posterior distribution. For ease of visualization, the posterior was arbitrarily partitioned into four distinctive groups. We emphasize the top-right group [B, F], which displays the most important signaling model parameter interactions. **[E-H]** First assessment of model parameter sensitivity. These panels show the same components as in [A-D] but the posterior is now conditioned on the maximum-a-posteriori probability (MAP) estimate of the model parameters.
(PDF)

## Acknowledgments

The successful completion of this research project owes much to the collaboration and support of esteemed colleagues and collaborators. We extend our deepest gratitude to Sabine Fischer, Tim Liebisch, Franziska Matthäus, and Simon Schardt for their pivotal roles in fostering insightful discussions and providing constructive feedback.

We also express our appreciation to Roberto Covino and his lab for their invaluable guidance and expertise throughout the project, especially for pointing us in the direction of the powerful SBI framework. Their thoughtful insights and constructive critiques have enriched the quality of our work.

Furthermore, we would like to acknowledge the Center for Scientific Computing (CSC) at Goethe University Frankfurt for granting us access to the Goethe-HLR cluster, which has been instrumental in facilitating the progress of this research.

## Author Contributions

**Conceptualization:** Michael Alexander Ramirez Sierra, Thomas R. Sokolowski.

**Data curation:** Michael Alexander Ramirez Sierra.

**Formal analysis:** Michael Alexander Ramirez Sierra.

**Funding acquisition:** Thomas R. Sokolowski.

**Investigation:** Michael Alexander Ramirez Sierra.

**Methodology:** Michael Alexander Ramirez Sierra, Thomas R. Sokolowski.

**Project administration:** Thomas R. Sokolowski.

**Resources:** Thomas R. Sokolowski.

**Software:** Michael Alexander Ramirez Sierra.

**Supervision:** Thomas R. Sokolowski.

**Validation:** Michael Alexander Ramirez Sierra, Thomas R. Sokolowski.

**Visualization:** Michael Alexander Ramirez Sierra.

**Writing – original draft:** Michael Alexander Ramirez Sierra.

**Writing – review & editing:** Michael Alexander Ramirez Sierra, Thomas R. Sokolowski.

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
