## [Decision Letter · Decision Letter 0]

3 May 2024

Dear Mr. Ramirez Sierra,

Thank you very much for submitting your manuscript "AI-powered simulation-based inference of a genuinely spatial-stochastic model of early mouse embryogenesis" for consideration at PLOS Computational Biology.

As with all papers reviewed by the journal, your manuscript was reviewed by members of the editorial board and by several independent reviewers. In light of the reviews (below this email), we would like to invite the resubmission of a significantly-revised version that takes into account the reviewers' comments.

Comments from Guest Editor:

The manuscript has been reviewed by two experts, and both cite a number of important concerns to be addressed. In short, the manuscript's alignment with publication priorities of PLoS Comp Biol need greater emphasis in the added value of the model for practical, translational goals. The reviewers both point out that the biology of this system has been well-investigated, but that the scientific advances opened by this computer model must be more clearly articulated. A substantial revision of this manuscript will be necessary to address the reviewer comments.

We cannot make any decision about publication until we have seen the revised manuscript and your response to the reviewers' comments. Your revised manuscript is also likely to be sent to reviewers for further evaluation.

Sincerely,

Thomas B Knudsen, PhD

Guest Editor

PLOS Computational Biology

Mark Alber

Section Editor

PLOS Computational Biology

Comments from Guest Editor:

The manuscript has been reviewed by two experts, and both cite a number of important concerns to be addressed. In short, the manuscript's alignment with publication priorities of PLoS Comp Biol need greater emphasis in the added value of the model for practical, translational goals. The reviewers both point out that the biology of this system has been well-investigated, but that the scientific advances opened by this computer model must be more clearly articulated. A substantial revision of this manuscript will be necessary to address the reviewer comments.

Reviewer's Responses to Questions

**Comments to the Authors:**

Reviewer #1: SUMMARY

Ramirez-Sierra and Sokolowski present an AI-powered model of inner cell mass differentiation during preimplantation development investigating, in particular, how this stochastic system enables an incredibly robust output. They manage to reproduce many of the known properties of the ICM, including the early plasticity, the critical window of FGF segregation to allow for intercellular communication, and the fascinating robustness even when scaling the ICM, to mention a few. It’s an extensive computational-heavy manuscript of a highly studied system, as the long reference list also indicates. The manuscript does not add new experimental data. Unfortunately, the importance to the field and the biological significance are unclear, and new experimentally testable predictions are lacking. Based on these concerns, I unfortunately cannot recommend publication in PLOS Computational Biology.

CONCERNS

1. As stated in the journal’s aim: “PLOS Computational Biology publishes original research that clearly demonstrates novelty, importance to a particular field, biological significance, and conclusions that are justified by the study”. I can positively argue for the novelty of this study: Applying AI technology to the ICM has, to my knowledge, never been done before. Also, the conclusion (lines 697-699) is backed up by the manuscript. However, it’s unclear how this study provides “unique insights” (line 698) and “deepen our understanding of the developing early mouse embryo” (line 700). All the five suggestions listed in lines 117-124 are already known. These are not new findings. Therefore, the importance to the field and its biological significance are highly unclear.

2. This study presents a versatile computational framework, but no new experimentally testable predictions are suggested. To provide value to the community, this tool should open up one or more new directions with testable predictions that have not already been carried out. What new experiments would the authors suggest? It’s challenging because the ICM is already a highly studied system. Still, as the authors note on lines 701-702, the proposed framework may be used “for exploring similar controlled stochastic processes in related biological systems.” The question is, which systems do the authors have in mind? And how, more precisely, do the authors imagine the presented tool can provide new, currently unavailable insights?

3. The authors emphasize several times the spatial-stochastic capabilities of their model (it’s even part of the manuscript title, and line 97 says “under truly stochastic conditions”). Still, they present a static 2D model. ICM cells (and cells in general) are highly 3D. They move around, divide, and change how many cell neighbors they have, the distance to them, and who their neighbors are. This introduces a whole new level of spatial-stochasticity that has been captured before (as far back as refs. 2 or 11). This undermines the selling point of this manuscript. Can the model presented here work in a dynamically changing 3D world to capture and build predictions for other systems than what can be carried out in a dish?

4. I tried to install and run the GitHub code provided. The scripts are pedagogically named with figure numbers. Unfortunately, I failed. Apparently, the current version of SciPy isn’t working, so I downgraded to the version listed in the requirement file provided. Even then, by testing Fig6.py and Fig8.py, I get a “min() arg is an empty sequence” ValueError. I’m likely making a simple mistake. That said, please provide better guidance on how to run the presented scripts and reproduce the figures. Please provide the remaining analysis data files necessary to check the study's validity.

5. The model is very complicated and not particularly minimal, with 24 fixed parameters (Table 3) and 19 free parameters (Table 4). Are all these parameters really needed for what the authors want to communicate? Can Table 4 be used to make experimentally testable predictions?

6. Why is AI needed for what the authors try to achieve? Please elaborate.

7. What boundary conditions are the authors applying to the 2D grid of cells?

8. Lines 19 and 122 (statement 4) are potentially exciting and new. Please elaborate on those.

Reviewer #2: GENERAL COMMENTS:

This exhaustive treatise provides a mathematical-computational reconstruction of the 2nd lineage specification in mouse, based on the NANOG-GATA6-FGF4 signaling axis that specifies the distribution of pluripotent ICM cells into epiblast (EPI) versus primitive endoderm (PrE) lineages. The model essentially mirrors the self-organization and patterning that has been well described in the literature (eg, Zernicka-Goetz, Rossant, and Tam labs) and more recently characterized by single cell RNAseq profiling.

1. Problem formulation: FGF4 signaling through FGFR1, ERK, and FOXO (not mentioned) is a key element of this GRN, and in this manuscript provides essentially a biological framework for multiscale modeling and simulation. It would be useful to provide some background into the translational value of modeling this period in mouse (E2.5 - E4.5) and human development. The work is a seminal piece in terms of technological achievement, but from a practical perspective what is the need for mechanistic understanding of developmental liabilities during this period? For example, do adverse outcomes show up as peri-implantation failure, developmental incompetence, vulnerability to developmental toxicants, pregnancy complications? Does it offer value for improving IVF? Does it offer value in engineering 3D microsystems (blastuloids, gastruloids)? Can it advance scientific understanding of in vitro embryonic stem (ECS) or induced pluripotent stem cell (iPSC) models, particularly with regards to naive, primed, determined states of ESC differentiation, developmental programming, and canalization?

2. The computational framework is a 2D lattice where individual cells are represented as a 'voxel' (I thought voxel referred to volume units; please explain the computational representation a little better, and whether it is truly a voxel dimension or multi-pixels. Also, can they please comment on why a 2D lattice was used to represent a 3D ICM (computational cost?).

3. The SNPE algorithm provides a stochastic simulator to parameterize a desired system behavior. This is really interesting and they might relate the advantages of this approach to the evolutionary algorithm used to stochastically model shape homeostasis in a computer 'virtual embryo' model of early development (Andersen et al. (2009) https://pubmed.ncbi.nlm.nih.gov/19199386/ or other kinds of morphogenetic algorithms.

4. A general comment on verbal quality. The text is detailed and clearly written, but the large numbers of acronyms make the reading a bit cumbersome, especially the 'comparative analysis' section. I realize the practical value in defining technical terms in this already lengthy manuscript (50 pages of single line text); however, it is distracting that several abbreviations (eg, ITWT, RTM) are defined multiple times in the manuscript. Please check these and other cases of acronyms and define where needed. One example of an appropriate re-definition is in the legend for Figure 2.

5. The Methods section has several important tables on parameters at the molecular, cell and tissue levels; however, it would be helpful to add a new table to the Results (circa Line 1342) that synopsizes the dry experiment scenarios.

6. Discussion: the opening 3 paragraphs of the Discussion are essentially rehashing introductory information (along with redefining acronyms) that could instead be part of a problem formulation statement. I would recommend that the Discussion opens with a paragraph that summarizes the key findings in the order that they will be discussed.

SPECIFIC COMMENTS:

1. The paper is based on mouse, but the first few paragraphs of the Introduction do not explicitly state this even though they define E2.5 - E4.5. In so doing, they should also define 'Embryonic day (E)' as used in the timeline.

2. Figure 4 and corresponding text: the simulated expression profiles in Figs 4D, 4E, and 4F are impressive. The authors simply state that this "mirrors experimental findings" and cite [19, 30, 84, 88]. Please indicate which of those findings, if any, profiled cell clusters by single cell RNA-seq (sc-RNAseq), and if so can the authors add something more than merely citing these references to support their assertion of molecular recapitulation.

3. Lines 403-406: more context is needed to introduce the autocrine/paracrine modes before delving into the computational details.

4. Figures 6 and 7 legends: please define the colors used in the graphs.

5. Please add discussion on the mouse to human extrapolation value of the models.

6. I am not quite clear on what the authors mean in the Methods by 'biophysically-realistic mechanistic modeling for quantitative understanding. It does not seem to reflect cellular biomechanics, but rather cellular trajectories over time. Terminology along those lines in the parlance of modern scRNAseq or spatial transcriptomic studies is 'pseudotime' reconstruction. Please explain terminology, if appropriate.

**Have the authors made all data and (if applicable) computational code underlying the findings in their manuscript fully available?**

Reviewer #1: **No: **As stated by the authors: "All remaining analysis data files will be available after acceptance." See also reviewer comment 4.

Reviewer #2: Yes

PLOS authors have the option to publish the peer review history of their article (what does this mean?). If published, this will include your full peer review and any attached files.

Reviewer #1: No

Reviewer #2: No
---

## [Decision Letter · Decision Letter 1]

10 Sep 2024

Dear Mr. Ramirez Sierra,

We are pleased to inform you that your manuscript 'AI-powered simulation-based inference of a genuinely spatial-stochastic gene regulation model of early mouse embryogenesis' has been provisionally accepted for publication in PLOS Computational Biology. Reviewer 1 has a few remaining comments which I assume you can address along with any remaining formatting changes that might be required.

Best regards,

Christoph Kaleta

Section Editor

PLOS Computational Biology

Christoph Kaleta

Section Editor

PLOS Computational Biology

Reviewer's Responses to Questions

**Comments to the Authors:**

Reviewer #1: The authors have done substantial revision to the manuscript. They have added several testable predictions, and better highlighted how their work compares to previously published computational models of the mouse blastocyst. I can recommend publication.

Two minor suggestions:

- The abstract reads more like a general introduction to the field with close to no new findings mentioned at all. I would suggest the authors to elaborate on the new findings and the experimentally testable predictions already in the abstract to emphasize how this manuscript moves the field forward.

- Figure 7D: Please state in the figure legend to this panel what the numbers indicate.

**Have the authors made all data and (if applicable) computational code underlying the findings in their manuscript fully available?**

Reviewer #1: **No: **Complete data bank missing.

PLOS authors have the option to publish the peer review history of their article (what does this mean?). If published, this will include your full peer review and any attached files.

Reviewer #1: No

---

## [Editor Report · Acceptance letter]

22 Oct 2024

PCOMPBIOL-D-24-00452R1 

AI-powered simulation-based inference of a genuinely spatial-stochastic gene regulation model of early mouse embryogenesis

Dear Dr Ramirez Sierra,

I am pleased to inform you that your manuscript has been formally accepted for publication in PLOS Computational Biology. Your manuscript is now with our production department and you will be notified of the publication date in due course.

With kind regards,

Anita Estes
